# INSTANTIR: BLIND IMAGE RESTORATION WITH INSTANT GENERATIVE REFERENCE

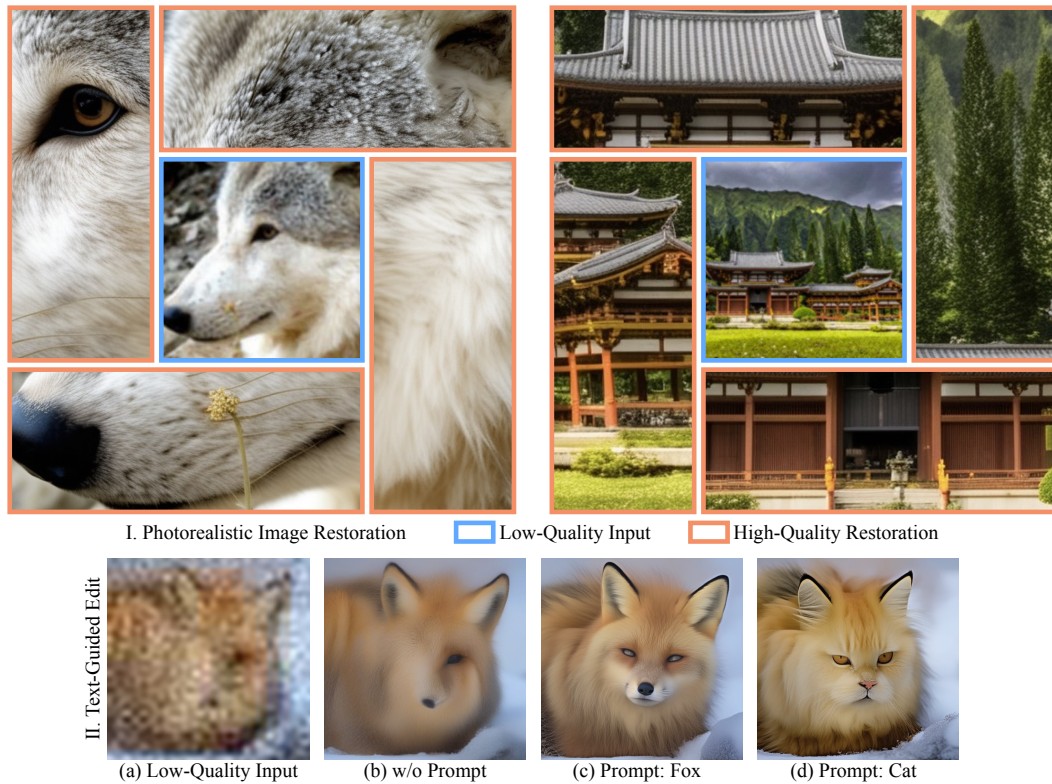

I. Photorealistic Image Restoration ☐ Low-Quality Input ☐ High-Quality Restoration

II. Text-Guided Edit

(a) Low-Quality Input  (b) w/o Prompt  (c) Prompt: Fox  (d) Prompt: Cat

Figure 1: I. INSTANTIR presents exceptional capability in reproducing photorealistic details. II. INSTANTIR provides an active interface for natural language guidance, helps handling large degradation and features creative restoration with semantic editing.

## ABSTRACT

Handling test-time unknown degradation is the major challenge in Blind Image Restoration (BIR), necessitating high model generalization. An effective strategy is to incorporate prior knowledge, either from human input or generative model. In this paper, we introduce Instant-reference Image Restoration (INSTANTIR), a novel diffusion-based BIR method which dynamically adjusts generation condition during inference. We first extract a compact representation of the input via a pre-trained vision encoder. At each generation step, this representation is used to decode current diffusion latent and instantiate it in the generative prior. The degraded image is then encoded with this reference, providing robust generation condition. We observe the variance of generative references fluctuate with degradation intensity, which we further leverage as an indicator for developing a sampling algorithm adaptive to input quality. Extensive experiments demonstrate INSTANTIR achieves competitive performance and offering outstanding visual quality. Through modulating generative references with textual description, INSTANTIR can restore extreme degradation and additionally feature creative restoration.

# 1 INTRODUCTION

Image restoration seeks to recover High-Quality (HQ) visual details from Low-Quality (LQ) images. This technology has a wide range of important applications. It can enhance social media contents to improve user experience (Chao et al., 2023). It also functions at the heart in industries like autonomous driving (Patil et al., 2023) and robotics (Porav et al., 2019) by improving adaptability in diverse environments, as well as assists object detector in adverse conditions (Sun et al., 2022).

Image restoration remains a long-standing challenge extending beyond its practical application. The information loss during degradation makes a single LQ image corresponds to multiple plausible restorations. This ill-posed problem is further exacerbated in Blind Image Restoration (BIR), where models are tested under unknown degradation. A common strategy is to leverage prior knowledge. Reference-IR models use other HQ images to modulate LQ features, requiring additional inputs with similar contents but richer visual details (Lu et al., 2021). Generative approaches, on the other hand, directly learn the HQ image distribution. The input is first encoded into a hidden variables $z$, which servers as the generation condition to sample HQ image from the learned distribution $p(y|z)$. Although generative methods achieve single-image restoration, they are prone to hallucinations that produce artifacts in restoration (Yang et al., 2020). This happens when the encoder fails to retrieve accurate hidden variable due to the input distribution shift in degradation. Existing methods improve robustness by training on more diverse synthetic degradation data or introduce discrete feature codebook. We argue that these are only shot-term solutions. Alternative methods are pendding to be explored to better address unknown inputs in BIR.

In this paper, we present INSTANTIR, a dynamic restoration pipeline that iteratively refines generation condition using a pre-trained Diffusion Probabilistic Model (DPM). INSTANTIR employs two complementary way for processing input LQ image. First, a pre-trained vision encoder extracts compact representation from degraded content. The encoder's high compression rate enhances the robustness in the extracted representation, while retaining only high-level semantics and structural information. Next, we introduce the *Previewer* module, a distilled DPM capable of one-step generation. At each generation step, the previewer decodes current diffusion latent using the compact representation, providing a restoration preview resembles original input in high-level features. This preview serves as an instant generative reference to guide the *Aggregator* in encoding identity and other fine-grained missing from the compact representation. We observe in experiments that the previewer tends to decode aggressively when the input is clear, resulting in high variance in restoration previews. We take this as a reliable indicator of input image quality, and develop an adaptive sampling algorithm that amplifies the fine-grained encoding with relatively high quality inputs. Additionally, we find the previewer is controllable through text prompts, which produces diverse generative references and enables semantic editing with restoration. Our contributions are as follows:

1. We explore a novel BIR method that iteratively aligns with the generative prior to address unknown degradation;

2. We introduce a novel architecture based on pre-trained DPM, which dynamically adjusts the generation condition by previewing intermediate outputs;

3. We develop sampling algorithms tailored for our pipeline, enabling both adaptive and controllable restoration to text prompts;

4. We perform extensive evaluations to validate the effectiveness of the proposed methods.

# 2 RELATED WORK

## 2.1 DIFFUSION MODEL

DPM is a class of generative model that generate data by iteratively denoising from Gaussian noise (Sohl-Dickstein et al., 2015; Ho et al., 2020; Song et al., 2020b). Typically, a neural network with a UNet architecture (Ronneberger et al., 2015) is trained to predict the noise added at each inference step. DPM offers superior mode coverage compared to Variational Autoencoders (VAE) (Kingma & Welling, 2013) and outperform GAN-based models (Goodfellow et al., 2020) in generation quality without the need of adversarial training (Dhariwal & Nichol, 2021). These advantages establish DPM as the leading approach in vision generative models. By incorporating

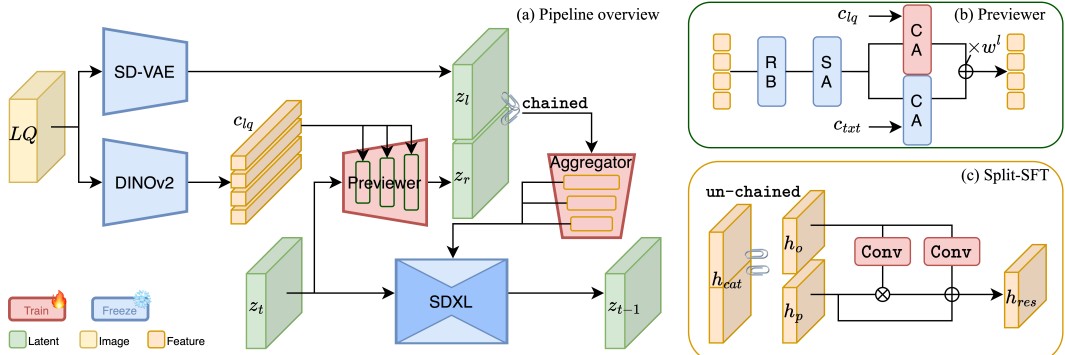

Figure 2: (a) Overview of the INSTANTIR pipeline. INSTANTIR utilizes two pre-trained encoder for processing LQ image at different levels. DINOv2 extracts compact representation $c_{lq}$ robust to degradations, providing high-level guidance for sampling the generative reference $Z_r$ from the refined posterior $p(z_0|z_t, c_{lq})$. SDXL's VAE encodes the LQ latent $Z_l$, preserving fine-grained details. (b) A Previewer model block. RB denotes Residual-Block and SA/CA corresponds to Self-Attention/Cross-Attention. We introduce a new CA to process the two modalities in parallel, the output is regulated by a hyperparameter $w^l$. (c) Connector between the Aggregator and SDXL. $Z_r$ and $Z_l$ are spatially concatenated in the Aggregator to minimize additional parameters channel-wise. Finally, the outputs from the Aggregator are split and fused using Spatial Feature Transform.

additional inputs, DPMs can learn diverse conditional distributions (Nichol & Dhariwal, 2021), with the most widely used application being text-to-image (T2I) generation (Rombach et al., 2022; Saharia et al., 2022a; Ramesh et al., 2022). Leveraging the flexibility of text inputs and the vast amount of text-image training data (Schuhmann et al., 2022), these models are capable of generating images with exceptional visual quality and remarkable diversity, forming the foundation for many subsequent excellent work in vision generative models (Wang et al., 2024c;a).

## 2.2 BLIND IMAGE RESTORATION

The task setting makes BIR particular valuable in real-world applications. The major challenge in BIR is the input distribution gap between training and testing data. Previous work have explored multiple ways to address this issue. Feature quantification is widely used in generative-based methods (Esser et al., 2021; Van Den Oord et al., 2017; Zhou et al., 2022). They align the encoded LQ image features to a learnable feature codebook, ensuring the input to generator is unaffected by domain shifts. However, this hard alignment constraints the generation diversity and quality by the capacity of the discrete codebook. Previous work have also explored the application of powerful DPM in BIR. Some approaches design specialized architectures and train DPMs from scratch (Saharia et al., 2022b; Sahak et al., 2023; Li et al., 2022), while the others apply additional modules on pre-trained T2I model (Wang et al., 2024b; Yu et al., 2024; Sun et al., 2024a), leveraging their large-scale prior. In many practical scenarios, HQ images with similar contents, such as those from photo albums or video frames, are available. This has spurred interest in restoring images using reference-based methods (Cao et al., 2022; Jiang et al., 2021; Lu et al., 2021; Xia et al., 2022; Yang et al., 2020; Zhang et al., 2019). They adopt regression models to learn how to transfer high-quality features to LQ images, enhancing details restoration.

## 3 METHODOLOGY

The distribution gap between training and testing data exacerbates the ill-posed nature of BIR, causing hallucinations in generation-based IR models and producing artifacts. We attribute this to the error in encoding LQ image, and propose a generative restoration pipeline that refines the LQ encodings with generative references. This is achieved by exploiting the reverse process of DPM. Specifically, we first encode the LQ image into a compact representation via pre-trained vision encoder, capturing global structure and semantics to initiate diffusion generation. Conditioned on this embedding, our Previewer module generates a restoration preview at each diffusion time-step. The

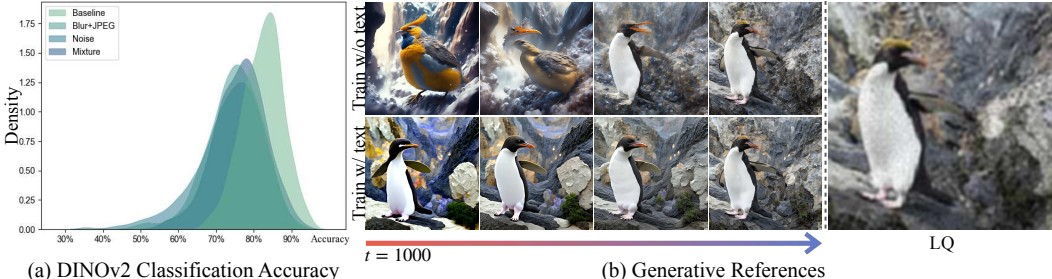

(a) DINOv2 Classification Accuracy                    (b) Generative References

Figure 3: (a) Zero-shot classification accuracies of DINOv2 on ImageNet-1K under various degradations, showing the robustness of its representations. (b) Sampling from the refined posterior $p(z_0|z_t, c_{lq})$ across diffusion time-steps. Generative references conditioned on $c_{lq}$ resemble the LQ input on high-level features and gradually converge toward the target mode in the reverse process.

preview resembles to the input image with more plausible details, and they are further fused in the Aggregator module to preserve fidelity. Finally, the adjusted LQ encoding is used to control the pre-trained DPM for a fine-grained diffusion step.

### 3.1 PRELIMINARIES

DPM involves two stochastic processes named forward and reverse process (Ho et al., 2020). In the forward process, *i.i.d.* Gaussian noise is progressively added to the image $\boldsymbol{x}$. The marginal distribution of diffusion latent $\boldsymbol{x}_t$ follows $\mathcal{N}\left(\alpha_t\boldsymbol{x}, \beta_t\boldsymbol{I}\right)$, where $\alpha_t$ and $\beta_t$ are hyperparameters defining the forward process. $\boldsymbol{x}_t$ converges to pure noise as $t$ increases, and the reverse process generates images by inverting the forward process. Generally, we train a neural-network to predict the noise added at each time-step by minimizing the diffusion loss:

$$\mathcal{L}_{diff} = \mathbb{E}\left[\|\boldsymbol{\epsilon}_\theta\left(\boldsymbol{x}_t, t\right) - \boldsymbol{\epsilon}\|^2\right], \tag{1}$$

where $\boldsymbol{\epsilon}_\theta$ denotes the noise-prediction network. At each step in the reverse process, we can retrieve a denoising sample with the predicted noise and re-parameterization (Karras et al., 2022):

$$\hat{\boldsymbol{x}} = \frac{\boldsymbol{x}_t - \beta_t\boldsymbol{\epsilon}_\theta\left(\boldsymbol{x}_t, t\right)}{\alpha_t}. \tag{2}$$

In the open-sourced T2I model Stable Diffusion (SD) (Rombach et al., 2022), the noise-prediction network $\boldsymbol{\epsilon}_\theta$ is additionally conditioned on a text input that describes the target image. Moreover, SD employs a VAE to move the input $\boldsymbol{x}_t$ into latent space $\boldsymbol{z}_t$, compressing inputs by a factor of 48 and significantly reduces the memory usage to enable image generation up to $512^2$ resolution.

### 3.2 ARCHITECTURE

The restoration pipeline of INSTANTIR consists of three key modules: Degradation Content Perceptor (DCP) for compact LQ image encoding, Instant Restoration Previewer for generating references on-the-fly during the reverse process, and Latent Aggregator for integrating restoration references.

**Degradation Content Perceptor** Human visual perception can easily tell the meaning and subjects of images even when they are heavily degraded. The same thing happens to vision recognition models. In Fig. 3(a) we test the zero-shot classification accuracy of DINOv2 (Oquab et al., 2023) on ImageNet-1K (Deng et al., 2009) under various degradations including noise, blur and JPEG artifacts. DINOv2 sustains 80% accuracy even under a mixture of degradations. The high-level information in DINO's representation can provide semantic guidance for the reverse process, yielding samples closely resemble the LQ input in these features. We employ the compact representation extracted from pre-trained DINOv2, and modulated it by a learnable Resampler (Han et al., 2024). For the $l$-th cross-attention block, we introduce an additional cross-attention operation:

$$\boldsymbol{f}_{out}^l = \boldsymbol{f}_{in}^l + \text{CrossAttn}\left(\boldsymbol{f}_{in}^l, \boldsymbol{c}_{txt}\right) + w^l \cdot \text{CrossAttn}\left(\boldsymbol{f}_{in}^l, \Phi\left(\boldsymbol{c}_{lq}, t\right)\right), \tag{3}$$

where $\Phi$ denotes the DCP module and $\boldsymbol{c}_{lq}$ is the LQ context matrix. We retain the text cross-attention here as it is a crucial part of the pre-trained T2I model that synthesizes high-level semantics. Jointly

training DCP with textual transformation allows it to focus on low-level information absent in the other modality. We introduce a hyper-parameter $w^l$ to regulate their behaviors. Note that the DCP also takes time-step $t$ as input to establish temporal dependency in the output. Specifically, we use adaptive layer-normalization to modulate the context matrix from the DCP according to time-step $t$:

$$\Phi\left(\boldsymbol{x}, t\right) = \boldsymbol{\mathcal{T}}_{scale} \odot \texttt{LayerNorm}\left(\boldsymbol{c}_{lq}\right) + \boldsymbol{\mathcal{T}}_{shift}, \tag{4}$$

where, $\boldsymbol{\mathcal{T}}_{scale}, \boldsymbol{\mathcal{T}}_{shift}$ are calculated from the time-step. We train the DCP module on a frozen diffusion model using the standard diffusion loss in Eq. 1.

**Instant Restoration Previewer**   The compact representation encoded by the DCP, while robust against degradation, lacks low-level information. We introduce Previewer, a diffusion model generates from current diffusion latent instead of noise, to decode generative references from the DCP encoding. Decoding at each diffusion time-step requires $(T\left(T+1\right)/2)$ network forward passes with the vanilla T2I model. To streamline this process, we fine-tune the Previewer using consistency distillation (Luo et al., 2023) to make it a one-step generator. For diffusion latent $\boldsymbol{z}_s$ at time-step $s$, we first obtain the Previewer output conditioned solely on $\boldsymbol{c}_{lq}$. Then, we perform a diffusion step using the pre-trained model from $\boldsymbol{z}_s$, conditioned on both $\boldsymbol{c}_{lq}$ and $\boldsymbol{c}_{txt}$, to reach $\boldsymbol{z}_t$. $\boldsymbol{z}_t$ is regarded as the ground-truth diffusion latent at time-step $t$ in the sampling trajectory. Finally, we get the preview of $\boldsymbol{z}_t$, again conditioned solely on $\boldsymbol{c}_{lq}$. The consistency distillation loss is then calculated by:

$$\mathcal{L}_{dist} = \|\Psi\left(\boldsymbol{z}_s, s, \Phi\left(\boldsymbol{c}_{lq}, s\right)\right) - \texttt{StopGrad}\left(\Psi\left(\boldsymbol{z}_t, t, \Phi\left(\boldsymbol{c}_{lq}, t\right)\right)\right)\|^2, \tag{5}$$

where $\Psi$ denotes the previewer model. Additionally, Eq. 5 trains the previewer to follow the sampling trajectory without $\boldsymbol{c}_{txt}$, removing its dependency on text conditions which are typically unavailable in BIR tasks. The consistency constraint (Song et al., 2023) of enforcing consistent outputs across time-step enabling the Previewer to decode generative references on-the-fly.

**Latent Aggregator**   The primary challenge in the BIR task is the input distribution shift. Previous work address this by aligning LQ features with reference HQ images or a learned feature codebook. The former takes extra inputs, while the latter is limited to a specific domain by the codebook capacity. In contrast, we generate reference features directly from diffusion prior. Since the compact embedding $\boldsymbol{c}_{lq}$ retains only high-level information, it is insufficient for the Previewer to reconstruct HQ images at larger time-steps, as shown in Fig. 3. Relying solely on reference preview incurs error accumulation, so the Aggregator anchors preview to the original input to prevent divergence in the reverse process. The input LQ image is encoded into SD's latent space and spatially concatenated with the preview. This expanded input remains compatible to the diffusion UNet, allowing the Aggregator to be initialized as a trainable copy of UNet compression path following (Zhang et al., 2023). We remove text cross-attention layers to make the Aggregator lightweight and independent of textual conditions like the Previewer. The preview and LQ hidden featrues are fused in the spatial-attention layers, which are further integrated via Spatial Feature Transform (SFT) (Wang et al., 2018). For hidden feature $\boldsymbol{H}^l$ at the $l$-th layer in the Aggregator, we first split it spatially into $\boldsymbol{h}_p^l$ and $\boldsymbol{h}_o^l$, corresponding to the hidden features of preview and LQ latent, and integrate them with SFT:

$$\boldsymbol{h}_{res}^l = \left(1 + \boldsymbol{\alpha}^l\right) \odot \boldsymbol{h}_p^l + \boldsymbol{\beta}^l; \boldsymbol{h}_p^l, \boldsymbol{h}_o^l = \texttt{Split}\left(\boldsymbol{H}^l\right), \tag{6}$$

where $\boldsymbol{\alpha}^l, \boldsymbol{\beta}^l = \mathcal{M}_\theta^l(\boldsymbol{h}_o^l)$ are two affine transformation parameters calculated from the feature map of LQ latent at this level. We extract multi-level features $\left\{\boldsymbol{h}_{res}^l\right\}_{l=1}^L$ from Aggregator using Eq. 6, and inject them into the corresponding part of U-Net expansion path through residual connections.

### 3.3   ADAPTIVE RESTORATION

INSTANTIR processes LQ image through two complementary ways: 1) extracting compact representation using the DCP, which is robust to degradation but loses fine-grained information; 2) encoding via the lossless SD-VAE and integrating with restoration preview, which is prone to errors in the SD-VAE. Under severe degradation, INSTANTIR may produce samples deviate from the target HQ image. In such cases, restoration previews exhibit small variation, suggesting the DCP struggles to provide guidance according to the input. We further analyze the trajectory of restoration previews during the reverse process, compare it with the denoising predictions from Eq. 2. We assess them on four degradation levels: HQ image, 4x downsampling, 8x downsampling and synthetic multi-degradation, representing decreasing input quality. Fig. 4 (a) illustrates the L2-distance between

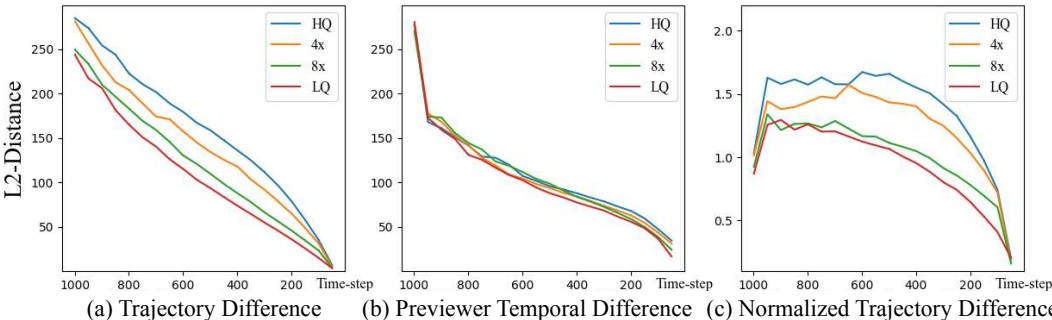

Figure 4: The evolution of the Previewer outputs during generation. (a) L2-distances between previews and denoising means; (b) temporal differences of the Previewer trajectory, measured by L2-distances between adjacent points; (c) relative distances between previews and denoising means.

these two trajectories, which increases monotonically as input quality improves. A pronounced disparity between preview and ordinary denoising prediction represents the Previewer is confident with the guidance, suggesting the input LQ image is informative. Based on this observation, we use the relative difference between two predictions as an indicator of input quality:

$$\delta = \frac{\|\Psi\left(z_t, t, \Phi\left(c_{lq}, t\right)\right) - \hat{z}_t\|^2}{\|\Psi\left(z_t, t, \Phi\left(c_{lq}, t\right)\right) - \Psi\left(z_{t+1}, t+1, \Phi\left(c_{lq}, t+1\right)\right)\|^2}, \tag{7}$$

where $\hat{z}_t$ is given by Eq. 2. From Fig. 4(b) we can see the Previewer is unstable at the beginning. The consistency training in Eq. 5 drives it to decode aggressively, causing large prediction variance during early reverse process where the input diffusion latent is too noisy. Normalizing the L2-distance between trajectories with Previewer's temporal difference effectively mitigates the temporal correlation as illustrated in Fig. 4(c). A larger $\delta$ indicates higher input quality, and the conditional signals from the Aggregator should be amplified to preserve fine-grained information from the original input. On the other hand, DPM is known to first generate low-frequency features such as global structure, and add high-frequency details in the later stage of the reverse process. A decreasing $\delta$ prevents INSTANTIR from divergence induced by generative references at the beginning. We provide pseudo-code of the proposed adaptive restoration (AdaRes) algorithm in Alg. 1. We provide more detailed discussion of the quality-fidelity trade off strategies in Appendix. B.

Surprisingly, although only the DCP module is explicitly trained on text-image data, IN-STANTIR demonstrates notable creativity following textual descriptions. By employing a text-guided Previewer, we can generate diverse restoration variations with compound semantics from both modalities. However, these variation samples can conflict with the original input, making them ineligible as generative references. We provide detailed analysis in Appendix. A. Inspired by previous work in image editing, we disable the Aggregator at later stage generation and let INSTANTIR renders semantic details according to LQ representation and

---

**Algorithm 1** Adaptive Restoration

**Input:** $\epsilon_\theta, \Psi, z_{lq}, c, \{\alpha_t, \beta_t | t = 1...T\}, \eta$
1: Sample $z_T \sim \mathcal{N}(\mathbf{0}, \beta_T \mathbf{I})$
2: Initialize $\bar{z}_{t+1}^{\Psi} = \mathbf{0}, z = \mathbf{0}, \delta = 1$
3: **for** $t$ in $[T, \ldots, 1]$ **do**
4: $\quad \bar{z}_t^{\Psi} = \Psi(z_t, t, c)$
5: $\quad z_{ref} = \bar{z}_t^{\Psi} + \delta \cdot \left(z_{lq} - \bar{z}_t^{\Psi}\right)$
6: $\quad \bar{z}_t = (z_t - \beta_t \epsilon_\theta(z_t, z_{ref}, t, c))/\alpha_t$
7: $\quad \delta = \|\bar{z}_t^{\Psi} - \bar{z}_t\|^2 \cdot \|\bar{z}_t^{\Psi} - \bar{z}_{t+1}^{\Psi}\|^{-2}$
8: $\quad z_{t-1} = (\beta_{t-1}/\beta_t)z_t - (\alpha_t/\beta_t - \alpha_{t-1})\bar{z}_t$
9: **end for**
**Output:** $z_0$

---

text prompt. This ensures the low-frequency features are succeeded from the Aggregator, meanwhile prevents the high-frequency semantics and noise from entering the final results.

## 4 EXPERIMENTS

### 4.1 IMPLEMENTATION DETAILS

INSTANTIR is built on SDXL (Podell et al., 2023) accompanied by a two-stage training strategy. In Stage-I, we train the Resampler in the DCP module connecting frozen DINOv2 and SDXL,

Table 1: Quantitative comparisons on both synthetic validation data and public real-world dataset. We highlight the best results in **bold** and the second best with underline.

| Dataset | Model | PSNR | SSIM | LPIPS | CLIPIQA | MANIQA | MUSIQ | NIQE |
|---------|-------|------|------|-------|---------|--------|-------|------|
| Synthetic | BSRGAN | 20.21 | 0.5214 | 0.7793 | 0.2072 | 0.2076 | 17.53 | 11.06 |
| | Real-ESRGAN | 19.92 | 0.5317 | 0.7554 | 0.2102 | 0.2331 | 17.39 | 9.840 |
| | StableSR | 20.42 | **0.5388** | 0.3751 | 0.4672 | 0.2602 | 52.33 | 5.274 |
| | CoSeR | 19.92 | 0.5114 | **0.3353** | **0.6651** | 0.4152 | 67.51 | **3.919** |
| | SUPIR | **20.46** | 0.4990 | 0.4090 | 0.4875 | 0.3081 | 56.43 | 4.408 |
| | INSTANTIR (ours) | 18.54 | 0.5126 | 0.3986 | 0.5497 | **0.4379** | **68.59** | 4.373 |
| Real-world | BSRGAN | 26.38 | 0.7651 | 0.4120 | 0.3151 | 0.2147 | 28.58 | 9.528 |
| | Real-ESRGAN | **27.29** | **0.7894** | 0.4173 | 0.2532 | 0.2398 | 25.66 | 8.561 |
| | StableSR | 26.40 | 0.7721 | **0.2597** | 0.4501 | 0.2947 | 48.79 | 7.724 |
| | CoSeR | 25.59 | 0.7402 | 0.2788 | **0.5809** | 0.3941 | 60.51 | 6.514 |
| | SUPIR | 26.41 | 0.7358 | 0.3639 | 0.3869 | 0.2721 | 42.72 | 8.550 |
| | INSTANTIR (ours) | 21.75 | 0.6766 | 0.3686 | 0.5401 | **0.4819** | **65.32** | **6.064** |

(a) Scenario 1: $512^2$ image restoration. The outputs of SUPIR and INSTANTIR are downsampled to $512^2$.

| Dataset | Model | PSNR | SSIM | LPIPS | CLIPIQA | MANIQA | MUSIQ | NIQE |
|---------|-------|------|------|-------|---------|--------|-------|------|
| Synthetic | BSRGAN | **21.32** | 0.5267 | 0.5611 | 0.4289 | 0.3299 | 37.97 | 9.566 |
| | Real-ESRGAN | 20.45 | 0.5202 | 0.5660 | 0.4566 | 0.3627 | 37.92 | 8.276 |
| | StableSR | 21.01 | **0.5490** | 0.3921 | 0.4526 | 0.2492 | 48.94 | 5.640 |
| | CoSeR | 20.50 | 0.5215 | **0.3488** | **0.6461** | 0.3939 | 64.84 | 4.265 |
| | SUPIR | 20.57 | 0.4569 | 0.4196 | 0.6286 | 0.3962 | 61.00 | 4.372 |
| | INSTANTIR (Ours) | 18.80 | 0.5076 | 0.3903 | 0.6111 | **0.4303** | **66.09** | **4.095** |
| Real-world | BSRGAN | **28.60** | 0.8141 | 0.3690 | 0.4720 | 0.2258 | 18.26 | 10.89 |
| | Real-ESRGAN | 28.13 | **0.8209** | 0.3647 | 0.4435 | 0.3229 | 35.31 | 10.16 |
| | StableSR | 27.79 | 0.8043 | **0.2514** | 0.4634 | 0.2901 | 46.54 | 7.475 |
| | CoSeR | 27.04 | 0.7683 | 0.2882 | **0.5847** | 0.4068 | 58.39 | **6.514** |
| | SUPIR | 26.10 | 0.5825 | 0.5429 | 0.4822 | 0.3232 | 44.95 | 9.582 |
| | INSTANTIR (Ours) | 21.89 | 0.6879 | 0.3601 | 0.5647 | **0.4389** | **62.58** | 8.024 |

(b) Scenario 2: $1024^2$ image restoration. We crop $512^2$ patches as inputs to 512-models and evaluate the quantitative metrics on the cropped area only.

followed by the Previewer's consistency distillation training (see Sec. 3.2). The Previewer is trained by applying Low-Rank Adaptation (LoRA) (Hu et al., 2021) on the base SDXL model for efficiency. By toggling the Previewer LoRA, we can seamlessly switch between the Previewer and SDXL, reducing memory footprint. After obtaining the DCP and Previewer LoRA, we proceed to Stage-II Aggregator training. The two-stage training ensures the Aggregator receives high-quality previews since the beginning of its training course.

We adopt SDXL's data preprocessing and conduct training on $1024^2$ resolution. In both two stages we use the AdamW (Loshchilov, 2017) optimizer with a learning rate of $1 \times 10^{-4}$. In Stage-I, we train the DCP module using a batch size of 256 over 200K steps, and distill the Previewer for another 30K steps with the same batch size. We train the Aggregator with a batch size of 96 over 200K steps in Stage-II. The entire training process spans approximately 9 days on 8 Nvidia H800 GPUs.

To enable Classifier-free Guidance (CFG) (Ho & Salimans, 2022) sampling, we apply LQ image dropout with a probability of 15% in both stages training. In all test experiments, we employ 30 steps DDIM sampling (Song et al., 2020a) with a CFG scale of 7.0.

## 4.2 EXPERIMENTAL CONFIGURATION

**Training Data** We synthesis LQ-HQ image pairs using Real-ESRGAN (Wang et al., 2021) with the default setting. As mentioned in Sec. 3.2, we conduct Stage-I training on the JourneyDB dataset (Sun et al., 2024b), a generated dataset with descriptive captions. While JourneyDB images are of extreme quality, they lack the textures in real-world images. Hence for Stage-II training, we incorporate publicly available texture-rich datasets to enhance model's ability to produce realistic

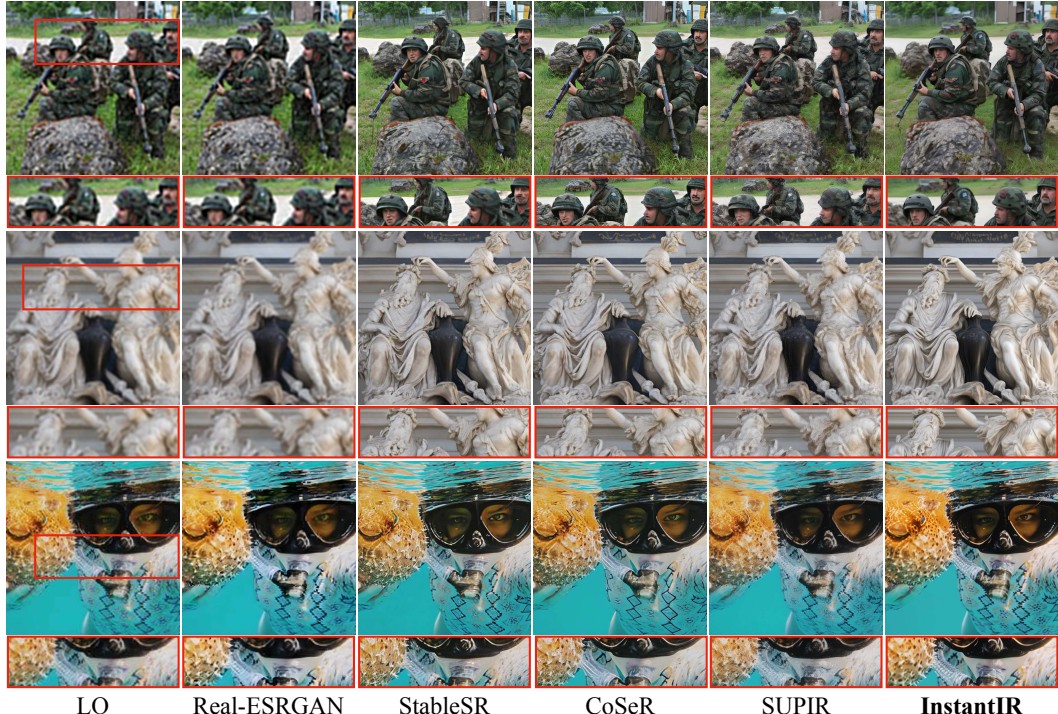

| LQ | Real-ESRGAN | StableSR | CoSeR | SUPIR | **InstantIR** |

Figure 5: Qualitative comparisons on real-world LQ images. Restorations from INSTANTIR are rich in details with global semantic consistency. Better viewed zoom in.

visual details. Specifically, we use DIV2K (Agustsson & Timofte, 2017), LSDIR (Li et al., 2023), Flickr2K (Timofte et al., 2017) and FFHQ (Karras et al., 2019).

**Test Setting** For a comprehensive evaluation, we test INSTANTIR on a synthetic dataset and public benchmarks following previous work. We synthesize $2,000$ multi-degradation samples from DIV2K and LSDIR validation sets using Real-ESRGAN pipeline, including deblur, denoise, SR and deJPEG simultaneously. We include a small portion of JourneyDB validation data to enhance benchmark diversity. We conduct evaluations on RealSR (Cai et al., 2019) and DRealSR (Wei et al., 2020) to assess model performance on real-world LQ images. We report full-reference metrics PSNR, SSIM, LPIPS (Zhang et al., 2018), if ground-truth targets are available, and non-reference metrics MANIQA (Yang et al., 2022), CLIPIQA (Wang et al., 2023), MUSIQ (Ke et al., 2021), and NIQE (Mittal et al., 2012) to quantitatively compare INSTANTIR with other models.

### 4.3 COMPARING TO EXISTING METHODS

We compare INSTANTIR with state-of-the-art models, including StableSR (Wang et al., 2024b), CoSeR (Sun et al., 2024a), SUPIR (Yu et al., 2024), BSRGAN (Zhang et al., 2021) and Real-ESRGAN (Wang et al., 2021). For the SD-based methods, we roughly balance the computational cost to 30 seconds per image on a V100 GPU. Since some of them are limited to $512^2$ resolution, we consider two test scenarios for a fair comparison: 1) models are tested on $512^2$ images with outputs of 1024-models scaled accordingly; 2) following SUPIR, the models are tested on $1024^2$ images by cropping $512^2$ patch as inputs to 512-models, metrics are evaluated on the cropped area only.

**Quantitative Comparison** The results are summarized in Tab. 1. INSTANTIR demonstrates superior image quality, as evaluated by an average ranking of **1.48** across non-reference metrics. IN-STANTIR continuously achieves the highest MUSIQ and MANIQA scores across all test settings, outperfoming the second best by large margins up to **22%** in MANIQA and **8%** in MUSIQ. Notably in scenario 1, despite halving the input data, INSTANTIR still performs comparably to SOTA models. While CoSeR achieves the best CLIPIQA scores closely followed by INSTANTIR, restorations

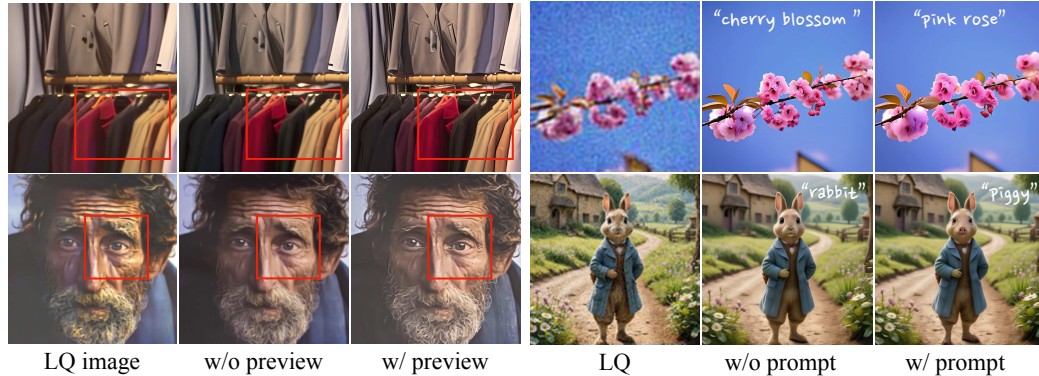

| LQ image | w/o preview | w/ preview | LQ | w/o prompt | w/ prompt |

(a) In-domain previews enhance detail restoration.    (b) Out-domain previews edits high-level semantics.

Figure 6: Visual examples of the previewing mechanism in INSTANTIR. Better viewed zoom in.

from 1024-models SUPIR and INSTANTIR are rich in details as shown in Fig. 5. We also observe the misalignment of PSNR and SSIM scores with visual quality as reported in the literature (Yu et al., 2024; Wang et al., 2024b). We include these metrics here for reference purpose.

**Qualitative Comparison** We provide some restoration samples on real-world LQ images in Fig. 5. Through leveraging the previewing mechanism, INSTANTIR actively aligns with generative prior, reducing hallucinations and producing sharp yet realistic details. In the second row of Fig. 5, while SUPIR's result contains rich textures, the absence of global semantic guidance causes the diver's body and mask to blend together. In contrast, the cognitive encoder in CoSeR helps it identifies statues in the second example. CoSeR employs a feature codebook to handle unknown degradations, which limits the generation of complex textures on the statues. Notably in the first row of Fig. 5, INSTANTIR is the only one that successfully recovers all four faces without distortion, suggesting its superior ability in capturing semantic and reproduce realistic details from diverse degradations.

## 4.4 ABLATION STUDY

**In-domain Reference for Detail Enhancement** Reference-based BIR models improve detail restoration by transferring high-quality textures from HQ references. INSTANTIR achieves this by querying the T2I model, eliminating additional inputs. To evaluate the effectiveness of generative references, we test INSTANTIR with different sources of reference. Specifically, we consider six reference sources with progressively increasing quality: the input LQ image, the target HQ image, DDIM mean from Eq. 2, unconditional restoration preview, restoration preview with DCP and additionally with text prompt. The latter three are both produced by our distilled Previewer. Results of this ablation study are summarized In Tab. 2a. Using the LQ image as reference yields the highest PSNR and SSIM value, as it preserves the maximum amount of original information. However, using the target HQ image will have these two metrics reduced. This occurs because INSTANTIR is designed to utilize dynamic generative references, and a fixed reference does not align with its training paradigm. We leave this limitation for future improvements. As more conditions are incorporated into the generative references, the restored image quality consistently increases, as indicated by perceptual metrics like CLIPIQA, despite decreasing PSNR and SSIM values. This observation aligns with the 'perception-distortion tradeoff' (Blau & Michaeli, 2018) that better perceptual quality comes at a price of worse distortion. We provide some visual samples in Fig. 6a.

**Out-domain Reference for Creative Restoration** Thanks to the efficiency of our Aggregator in processing latent inputs, INSTANTIR is able to perform high-level semantic editing during restoration, altering specific attributes of the subject and leaving other visual details unchanged as shown in Fig. 6b. We empirically find INSTANTIR offers better text-editing ability under heavy degradation where there is a relatively large information loss in the DINO representation. Detailed analysis as well as more visual samples are provided in Appendix. A.

Table 2: Ablation studies. The best results are highlighted in **bold**.

| Reference | PSNR | SSIM | LPIPS | CLIPIQA | MANIQA | MUSIQ | NIQE |
|---|---|---|---|---|---|---|---|
| LQ Image | 21.36 | 0.6417 | 0.4950 | 0.2415 | 0.2025 | 33.39 | 8.049 |
| HQ Image | 16.86 | 0.5791 | 0.3728 | 0.5078 | 0.3892 | 65.72 | 5.139 |
| DDIM Mean | 21.10 | 0.6066 | 0.4000 | 0.4515 | 0.3727 | 60.93 | 5.819 |
| Restoration Preview | 20.94 | 0.6108 | **0.3787** | 0.5023 | 0.4052 | 65.71 | 5.168 |
| +DCP | 18.77 | 0.5514 | 0.3933 | 0.5941 | 0.4687 | 70.45 | **4.658** |
| +DCP +prompt | 18.01 | 0.5202 | 0.4065 | **0.6489** | **0.5112** | **72.32** | 4.669 |
| Diffusion Latent | **23.07** | **0.7312** | 0.3830 | 0.3767 | 0.2924 | 49.23 | 4.894 |

(a) Ablation study of different reference types.

| AdaIN | AdaRes | PSNR | SSIM | LPIPS | CLIPIQA | MANIQA | MUSIQ | NIQE |
|---|---|---|---|---|---|---|---|---|
| ✗ | ✗ | 22.40 | 0.6937 | 0.3625 | 0.5361 | 0.4673 | 63.55 | 7.577 |
| ✗ | ✓ | 21.75 | 0.6766 | 0.3686 | **0.5401** | **0.4819** | **65.32** | **6.064** |
| ✓ | ✗ | **25.16** | **0.7247** | **0.3469** | 0.5188 | 0.4575 | 63.56 | 7.978 |
| ✓ | ✓ | 24.51 | 0.7102 | 0.3558 | 0.5319 | 0.4672 | 64.56 | 7.997 |

(b) Ablation study of the adaIN and AdaRes sampling.

**Adaptive Restoration**   Alg. 1 enhances restoration quality by gradually relaxing the constraints, which, however, incurs worse distortion. As shown in the first two rows of Tab. 2b, image quality scores increase as full-reference metrics degraded. On the other hand, diffusion model can occasionally exhibit color shift (Choi et al., 2022), where minor deviations in pixel values can significantly affect full-reference metrics. To address this issue, (Wang et al., 2024b) proposed normalizing generation outputs with color statistics derived from the LQ image, a post-process trick referred to as adaIN. We conduct an ablation study of Alg. 1 combined with adaIN in Tab. 2b. While adaIN can substantially improve full-reference metrics, it compromises image quality. Therefore, we opt not to incorporate this technique in INSTANTIR.

**Fresh Noise to Restoration Previews**   We additionally train an Aggregator that injects fresh noise to reference latents according to diffusion time-step. The noisy preview latent follows the same distribution as current diffusion latent, making the overall pipeline resemble a ControlNet model (Zhang et al., 2023). As shown in the last row of Tab. 2a, INSTANTIR significantly outperforms ControlNet with LQ image as conditional inputs. This highlights the effectiveness of the previewing mechanism in INSTANTIR for adjusting generation conditions during inference.

## 5 CONCLUSION

In this paper, we explore a novel method to address unknown degradations in BIR task. We first demonstrate the reliability of pre-trained DINOv2 in this low-level vision task, the extracted high-level representations are robust against degradations. Through exploiting the generation process of DPM, we propose to actively align with the generative prior to reduce the errors in encoding conditions. Our pipeline is implemented based on pre-trained SDXL model, referred to as INSTANTIR. Extensive experiments demonstrate the exceptional restoration capability of INSTANTIR, delivering competitive performance in quantitative metrics and visual quality. However, we observe some disparity in full-reference metrics such as PSNR and SSIM compared to SOTA models, partly due to our AdaRes algorithm which relaxes the generation constraints to promote quality. Integrating INSTANTIR with an adaIN post-processing step can mitigate this issue with a compromised restoration quality, reflecting the perception-distortion tradeoff. Future work could explore approaches to further advance this Pareto frontier, such as improving the interaction between generative references and conditions, as well as refining the previewer to more constraint references. Another potential limitation of INSTANTIR is its generalization across other image modalities, which will require fine-tuning the Aggregator with DINOv2 replaced by domain-specific image recognition models.

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

## A  CREATIVE RESTORATION

Although the Stage-2 Aggregator training of INSTANTIR is not conducted on images paired with text captions, INSTANTIR demonstrates notable flexibility in responding to text prompts. The compact representation from DINOv2, while robust against degradations, inevitably loses original information to different extent. This information loss leaves space for the injection of high-level semantic from text modality. In the DCP module, the two cross-attention layers are combined additively, allowing text descriptions to complement or modify the high-level features absent in DINOv2's representation. To validate this, we synthesize LQ images from the ImageNet-1K validation set using the Real-ESRGAN degradation pipeline. These images are then categorized based on their DINOv2 classification scores. We test the creative restoration outputs across these samples, using text prompts that either semantically close to with or deviate from that in the LQ images. Fig. 7- 9 visualize the restoration outputs, showing results without text prompts, with semantically aligned and deviated prompts, respectively. Across the first two rows, we can see that the intermediate restoration previews are easily manipulated when the DINO's classification scores are low, regardless of whether the text prompts close to or deviate from LQ images. This is because a low classification scores imply the high-level information is either absent or ambiguous in the DINO representation, allowing text cross-attention to dominate the joint transformation. As DINO classification scores increase, high-level information becomes more prominent in the representation and text-editing flexibility gradually vanish. At moderate classification scores illustrated in the third rows, the two modalities exert a balanced influence, and semantic conflicts can result in unpleasant outcomes. Finally, at high classification scores where the semantic is clear in DINO representation shown in the last two rows. It is difficult to manipulate the French bulldog, even at large diffusion time-steps as the high-level information from DINO overwhelms the semantics.

## B  QUALITY-FIDELITY TRADE OFF

Balancing generative capacity and fidelity to the input LQ image is a crucial aspect of developing generative-based BIR models. Among the compared methods in Tab. 1, DiffBIR (Lin et al., 2023), SUPIR (Yu et al., 2024) and StableSR (Wang et al., 2024b) each implements unique sampling algorithm to approach quality-fidelity balancing.

As a core component of DiffBIR pipeline, a pre-trained IR module not only provides diffusion sampling conditions for ControlNet, but also is used to balance quality-fidelity. Similar to INSTANTIR, DiffBIR retrieves the DDIM mean $\bar{z}_t$ at each diffusion time-step $t$. This mean $\bar{z}_t$ is then decoded into pixel space using SD-VAE to obtain $\bar{x}_t$. This intermediate output is used to calculate mean-squared loss with the IR module output, which typically holds high PSNR but sub-optimal perceptual quality, and the gradient is back-propagated with respect to current latent $\bar{z}_t$ to get an update direction. Compared to INSTANTIR, our pipeline is more efficient in two aspects: 1) we save both memory and computation induced by a pre-processing model; 2) we directly process the restoration previews in latent space using the Aggregator, eliminating the computational cost involved in calling SD-VAE and gradient propagation at every sampling step.

StableSR adopts the Controllable Feature Wrapping (CFW) module to balance quality-fidelity. Specifically, the SD-VAE is tuned on LQ images. The encoder is optimized for degradation robustness, ensuring it generates latent from LQ image that close to the corresponding HQ image. On the other hand, residual connections from the LQ encoder features are added to the decoder for preserving input information. These residual connections can be regulate with a hyper-parameter CFW-scale between $[0.0, 1.0]$. A larger CFW-scale enhances the LQ features in the decoder and thus improve fidelity. Since StableSR is trained on SD-2-1, the provided VAE checkpoint is not compatible with the SDXL model in INSTANTIR. However, we believe integrating this strategy into INSTANTIR could potentially enhance the flexibility in quality-fidelity balancing.

The encoder of SD-VAE is also fine-turned for degradation robustness in SUPIR. Unlike StableSR, SUPIR does not adjust the decoder for applying CFW module. SUPIR utilizes the degradation robust encoder as an initial restoration $\hat{z}_{lq}$. At each diffusion step, the diffusion mean $\bar{z}_t$ is interpolated with $\hat{z}_{lq}$ using a time-dependent scaler $k_t = (t/T)^{\tau}$. Smaller $\tau$ corresponds to larger $k_t$, making the interpolated mean closer to $\hat{z}_{lq}$.

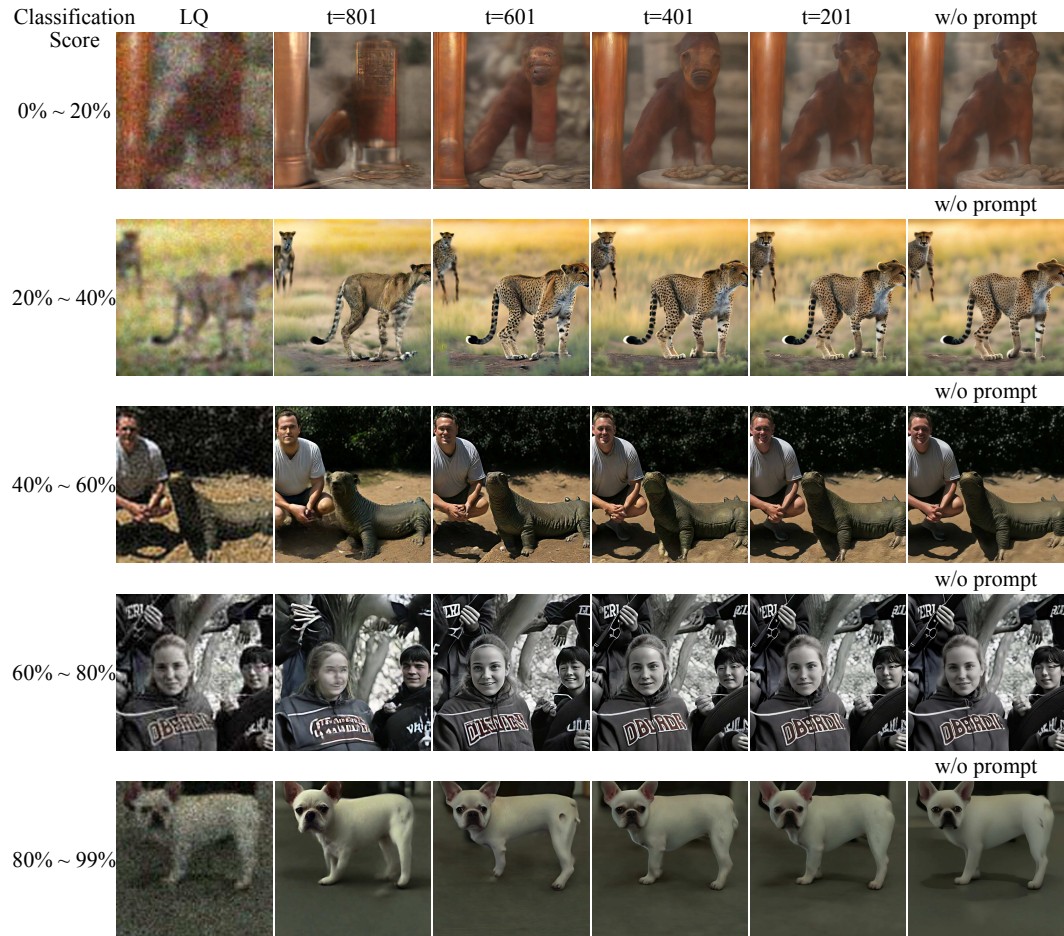

Figure 7: InstantIR outputs of synthesized LQ images from ImageNet-1K validation set. The images are categorized by DINOv2 classification scores. Column 2-5 visualize the generative references from the Previewer at different diffusion time-step.

In Alg. 1, the scaling factor $\delta$ is also time-dependent as $k_t$, which is beneficial for providing finer-grained control across time-steps. However, $k_t$ depends only on time-step $t$ while $\delta$ is adaptive to different inputs, offering additional flexibility. The idea of interpolation with $\hat{z}_{lq}$ in SUPIR's restoration-guided sampling algorithm is simple but effective. In Alg. 1, we borrow this idea and adapt it to INSTANTIR, where we interpolate the generative reference $\bar{z}_t^{\Psi}$ at each step with $\hat{z}_{lq}$. This interpolation prevent large distortion induced when previewing from a too noisy diffusion latent $z_t$.

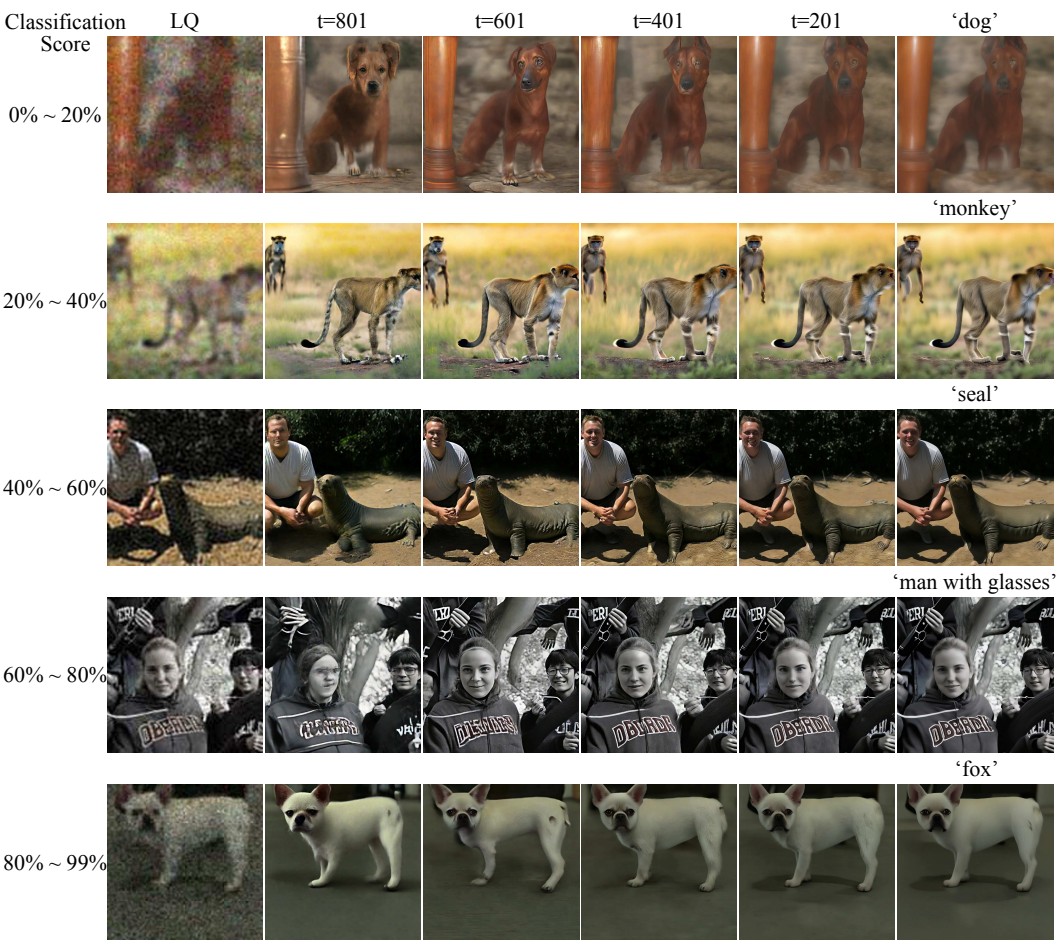

Figure 8: InstantIR outputs of synthesized LQ images from ImageNet-1K validation set, guided by semantically closed text prompts. The images are categorized by DINOv2 classification scores. Column 2-5 visualize the generative references from the Previewer at different diffusion time-step.

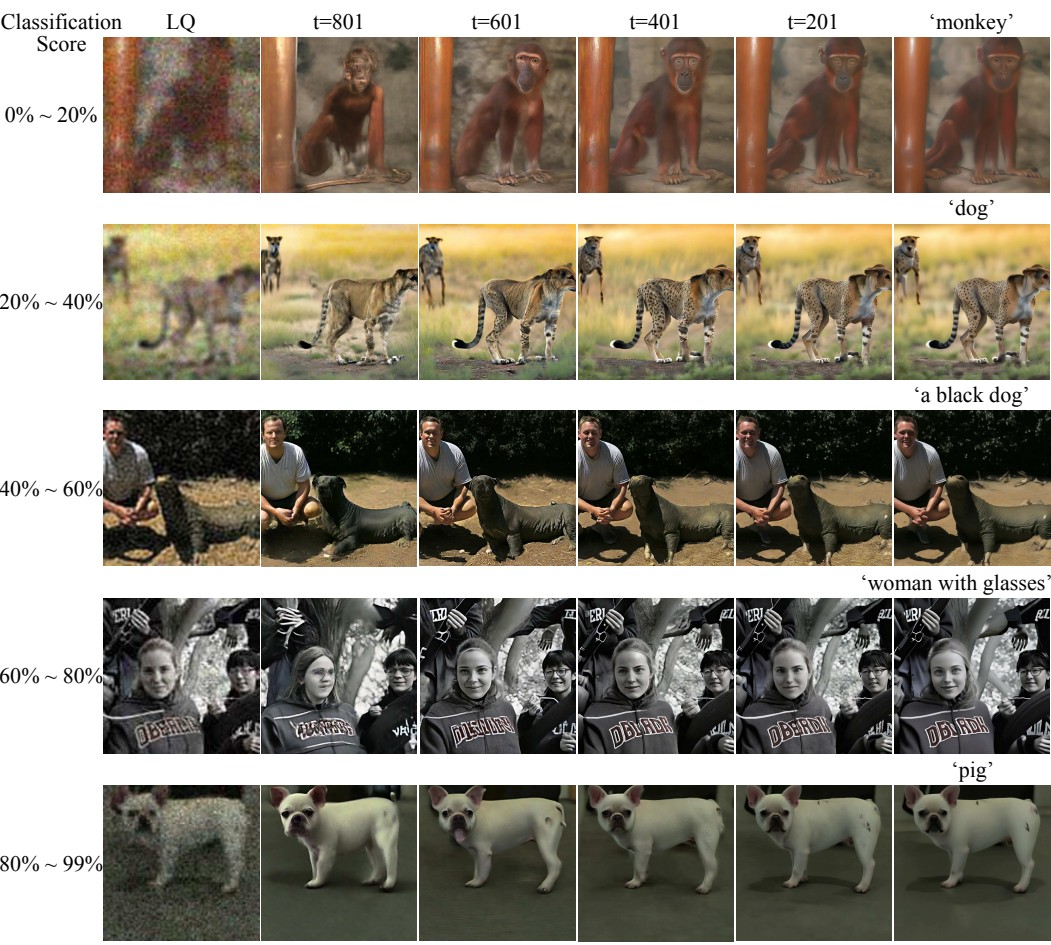

Figure 9: InstantIR outputs of synthesized LQ images from ImageNet-1K validation set, guided by semantically far text prompts. The images are categorized by DINOv2 classification scores. Column 2-5 visualize the generative references from the Previewer at different diffusion time-step.

