# OpenReview forum: "InstantIR: Blind Image Restoration with Instant Generative Reference"
_ICLR.cc/2025/Conference — Submitted to ICLR 2025_

### Official Review · Reviewer_KzYR · 2024-10-21

**Soundness:** 2
**Presentation:** 3
**Contribution:** 2
**Rating:** 5
**Confidence:** 5

**Summary:**

This paper proposes a new methodology for blind image restoration.
The main idea revolves around a new way to provide the degraded image as a condition to a pre-trained diffusion model (which is, to my knowledge, is typically done with a ControlNet).
Specifically, a pre-trained vision encoder first extracts a compact representation of the degraded image. This representation is combined with the diffusion state at each step, to form a preview of the generated image corresponding to that step (using a previewer). The previewed image is then used as the input condition for the diffusion model to produce the result in the next step.

**Strengths:**

1. The paper is overall clear and well written.
2. The proposed method is, to my knowledge, novel.
3. Connection to related work is good.
4. Experiments are thorough, including a convincing ablation study.

**Weaknesses:**

1. Compared to previous methods, the proposed method (InstantIR) seems to attain better perceptual quality, but this comes at the expense of worse distortion (e.g., PSNR, SSIM). Due to the perception-distortion tradeoff [1], this means that the proposed method cannot be claimed to be "SOTA", but just another point of the perception-distortion pareto frontier. Namely, to really be SOTA, a given algorithm needs to attain **better perceptual quality and better distortion simultaneously**, thus approaching the true perception-distortion tradeoff bound (dominating previous methods). Thus, it is not clear whether the proposed method is effective, because previous methods may also improve their perceptual quality at the expense of distortion (e.g., by tuning some hyper-parameters) and perhaps beat the proposed approach in terms of distortion at a given level of perceptual quality. Unfortunately, the authors do not discuss the perception-distortion tradeoff at all, which I believe is highly important given the fact that the proposed method's distortion is compromised in all data sets.
Note that the authors do briefly mention this limitation in the conclusion section (the compromised distortion compared to previous methods). But again, I believe that this should be thoroughly discussed, and it should be confirmed that the proposed approach is more effective than previous ones.

2. The conclusion/discussion section is quite short and limited. For example, a limitation I would specify is the design choice itself, which necessitates a degradation concept perceptor (e.g., DINO). What about other image modalities besides natural images (e.g., medical images, infra-red, compressed sensing), which DINO cannot handle? How can we then train InstantIR in such circumstances? Do we need to come up with a DINO-equivalent model to operate on images in other domains? My point is that InstantIR cannot be easily generalized/applied to other image modalities. I believe that this should be discussed, since many practitioners are working with other image modalities where blind image restoration is a crucial task.

3. While the proposed approach is novel, it is conceptually not that much different than previous methods (e.g., [2]). Specifically, the proposed method aims to generate samples from the posterior distribution (of natural images given a degraded image), which is an approach done in many previous works. Besides the empirical results, which, in my opinion, do not convince that the proposed approach is superior to previous ones (see point 1.), it is not clear why InstantIR offers any additional advantage over previous methods in terms of approximating the true sampler from the posterior.

4. The authors do not report NIQE scores for perceptual quality, which is standard in such works (e.g., [3]).

[1] Yochai Blau and Tomer Michaeli, The Perception-Distortion Tradeoff, CVPR 2018.

[2] Xinqi Lin et al., DiffBIR: Towards Blind Image Restoration with Generative Diffusion Prior, arXiv 2023

[3] Xintao Wang et al., Real-ESRGAN: Training Real-World Blind Super-Resolution with Pure Synthetic Data, ICCV 2021

**Questions:**

1. Could you please explain why the proposed method is superior to previous ones, specifically referring to the perception-distortion tradeoff? (see weakness 1). Why is it appropriate to say that InstantIR is SOTA, given the perception-distortion tradeoff? What evidence in the paper convinces that the proposed approach is more effective than previous ones? For example, if previous methods were trained slightly differently (with different hyper-parameters), they may attain the same perceptual quality as InstantIR, but with better distortion (e.g., PSNR, SSIM).

2. How can we apply/train InstantIR in other image modalities besides natural images?

---

> ### Author Response · Authors · 2024-11-21
> **Official Response (1/3)**
>
> Dear reviewer KzYR,
>
> We sincerely appreciate the reviewer's effort and constructive comments, especially **the recognition of our method’s novelty and its difference with previous ControlNet-based approaches**. Below, we will address your concerns one-by-one:
>
> > "Compared to previous methods, the proposed method (InstantIR) seems to attain better perceptual quality, but this comes at the expense of worse distortion (e.g., PSNR, SSIM)."
>
> Thank you for recognizing the perceptual quality delivered by InstantIR. This advantage is supported by the superior image quality metrics, including CLIPIQA, MANIQA, MUSIQ and the newly added NIQE as presented in both **Tab.1, Page 7**, and **Tab. 2, Page 10**.
>
> Regarding the distortion indicated by the declined PSNR and SSIM value, this can be attributed in part to our adaptive restoration algorithm (Alg. 1 on Page 6). **Alg. 1 improves image quality by progressively relaxing generation constraints**, which may result in **some distortions in fine details** when these features are being generated in the late diffusion process. However, this tradeoff prevents distorted low-level information from LQ images leaking into the outputs, thereby enhancing the overall perceptual experience, as illustrated in the first two rows of **Tab. 2(b), Page 10**.
> Moreover, diffusion models occasionally exhibit **color shifts** as reported in [1], which also **introduce subtle pixel-wise deviations**. Both of these two minor distortions can cause **declines in the full-reference metrics**.
>
> As a workaround, we can incorporate post-processing procedures like adaptive image normalization (adaIN) in [2]. We have added an additional ablation study of combining adaIN and Alg. 1 in **Tab. 2(b), Page 10**, as a direct comparison to the results in Tab. 1(a)'s Real-world benchmark. The **PSNR and SSIM can be adjusted to the same level as other methods, while maintaining the leading performance of InstantIR in image quality metrics**. While this adjustment can generalize across entries in Tab. 1, we **opt to retain the original results** since we prioritize human perspective quality in image restoration.
>
> InstantIR adjusted with color correction, best scores marked as **bold**:
> | Model | PSNR | SSIM | LPIPS | CLIPIQA | MANIQA | MUSIQ | NIQE |
> | ------- | ------- | ------ | ------- | ---------- | ---------- | --------- | ------ |
> | BSRGAN | 26.38 | 0.7651 | 0.4120 | 0.3151 | 0.2147 | 28.58 | 9.528 |
> | Real-ESRGAN | **27.29** | **0.7894** | 0.4173 | 0.2532 | 0.2398 | 25.66 | 8.561 |
> | StableSR | 26.40 | 0.7721 | **0.2597** | 0.4501 | 0.2947 | 48.79 | 7.724 |
> | CoSeR | 25.59 | 0.7402 | 0.2788 | **0.5809** | 0.3941 | 60.51 | 6.514 |
> | SUPIR | 26.41 | 0.7358 | 0.3639 | 0.3869 | 0.2721 | 42.72 | 8.550 |
> | **InstantIR (w/o adaIN)** | 21.75 | 0.6766 | 0.3686 | 0.5401 | **0.4819** | **65.32** | **6.064** |
> | **InstantIR (w/ adaIN)** | 24.51 | 0.7102 | 0.3558 | 0.5319 | 0.4672 | 64.56 | 7.997 |
>
>
> > "Due to the perception-distortion tradeoff [1], this means that the proposed method cannot be claimed to be "SOTA", but just another point of the perception-distortion pareto frontier."
>
> Thank you for raising this excellent point. Firstly, we believe that **improving human experience** should be the primary objective in developing BIR methods. As replied above, the **PSNR and SSIM can be adjusted to the same level as other methods, while maintaining the leading performance of InstantIR in image quality metrics**. But we opt to report the original results which is preferred by our visual evaluation.
> Secondly, InstantIR is claimed to be SOTA **not only because of these qualitative metrics but also because of its flexibility**, which allows users to tune the outputs and decide such tradeoff from their own preferences. This advantage comes from the reference mechanism in InstantIR. As detailed in **lines 463-477, Page 9**, distortions can be reduced by substituting generative references with original inputs; or increase image quality by using more semantic meaningful generative references. These two options can be mixed by adopting different types of references during the diffusion generation process, and this advantage cannot be quantitatively measured. We invite the reviewer to explore our local Gradio demo launching script presented in our anonymous Github: https://anonymous.4open.science/r/InstnatIR_anonymous-F4DF.

---

> > ### Comment · Reviewer_KzYR · 2024-11-22
> >
> > I highly appreciate the authors' efforts to address my review.
> >
> > Regarding the new results, it seems that InstantIR (w/ adaIN) is still worse than other methods on the perception-distortion plane (for some of the perceptual quality scores). For example, it achieves **both** a lower (worse) PSNR and a higher (worse) NIQE compared to CoSeR and StableSR. This is also true when comparing SSIM vs. NIQE. So I still don't see why InstantIR is **preferable** over previous methods (which do not require a reference image).
> >
> > I also don't see why "adjusting" the perceptual quality according to the user's experience (e.g., via a text prompt) is beneficial, if it results in a larger distortion than other methods. Let me try to explain this as follows. Suppose that we have the 2 versions of InstantIR noted above:
> >
> > 1. InstantIR-1: PSNR 24.51, NIQE 7.997, CLIPIQA 0.5319
> > 2. InstantIR-2: PSNR 21.75, NIQE 6.064, CLIPIQA 0.5401
> >
> > Now let's take StableSR and CoSeR:
> >
> > 1. StableSR: PSNR 26.40, NIQE 7.724, CLIPIQA 0.4501
> > 2. CoSeR: PSNR 25.59, NIQE 6.514, CLIPIQA 0.5809
> >
> > Now, suppose that some user would like the best possible perceptual quality, and he can choose to use either of these methods. In that case, the user would choose InstantIR-2 if NIQE is the desired measure, **or CoSeR if CLIPIQA is the desired measure**. Now, suppose that the user is interested in best distortion, but still wants a good level of perceptual quality. Then, clearly, StableSR would be the best choice, since it achieves the highest PSNR and the best perceptual quality according to both of these measures. I hope my explanation is clear enough.

---

> > > ### Author Response · Authors · 2024-11-23
> > > **Official Response (1/2)**
> > >
> > > Dear reviewer KzYR,
> > >
> > > We really appreciate the reviewer’s fast response and extra efforts devoted to our work. Below we will address the remaining concerns by showing that 1) while InstantIR may not overwhelmingly dominate, it shows **competitive performance** in quantitative evaluations; 2) the restoration results from InstantIR are **favored from human perspective**, according to the user study; 3) **the flexibility** to adjust the perception-distortion tradeoff at inference time is important to improve user experience.
> > >
> > > > “Regarding the new results, it seems that InstantIR (w/ adaIN) is still worse than other methods on the perception-distortion plane (for some of the perceptual quality scores). For example, it achieves both a lower (worse) PSNR and a higher (worse) NIQE compared to CoSeR and StableSR. This is also true when comparing SSIM vs. NIQE. So I still don't see why InstantIR is preferable over previous methods (which do not require a reference image).”
> > >
> > > Thank you for your thorough analysis of the quantitative comparisons. First of all, we would like to clarify that there isn't any reference image nor any additional inputs in InstantIR. We generate references from instant posterior as mentioned in our reply. If it is the computational cost that the reviewer is concerned about with ‘which do not require a reference image’, we have also clarified that all the results presented in Tab. 1 are generated using the same computational cost in **lines 421-423, Page 8**.
> > >
> > > The purpose of this supplementary result is to demonstrate that the distortion problem in InstantIR can be effectively mitigated via a simple post-processing step, producing a distortion level comparable to that of other methods. On the other hand, this result also highlights InstantIR's flexibility in being tuned on its own perception-distortion plane. In a word, we believe InstantIR offers **competitive overall performance** in quantitative evaluations.
> > >
> > > > “Now, suppose that some user would like the best possible perceptual quality, and he can choose to use either of these methods. In that case, the user would choose InstantIR-2 if NIQE is the desired measure, or CoSeR if CLIPIQA is the desired measure. Now, suppose that the user is interested in best distortion, but still wants a good level of perceptual quality. Then, clearly, StableSR would be the best choice, since it achieves the highest PSNR and the best perceptual quality according to both of these measures.”
> > >
> > > We appreciate the reviewer’s intuitive example. It appears that no single model holds a decisive advantage across all metrics, as each method demonstrates its own strengths depending on the metric used: StableSR wins the PSNR, CoSeR wins the CLIPIQA and InstantIR wins the NIQE. If we really have to make decision solely on a single metric, according to the result in the previous 'Official Response (1/3)', InstantIR would be preferred 43% of the time, compared to CoSeR and StableSR, which are each preferred 14% of the time. Nevertheless, we **disagree with relying on a single evaluation metric**, as it oversimplifies the complex tradeoffs inherent in image restoration. We believe this also aligns with the reviewer's perspective, that *InstantIR does not outperform all quantitative comparisons to be claimed as SOTA, which we respectfully acknowledge*.
> > >
> > > Meanwhile, it is worth noting that full-reference metrics, such as PSNR, often misalign with human perceptual quality, as widely discussed in [3,5,6]. In a word, while quantitative metrics offer valuable insights for comparing image restoration models, we argue that **human perceptual quality should be prioritized** to achieve a more comprehensive and realistic assessment. To this end, we conducted an anonymous user study during the author response period. This study involved 10 sets of real-world and digital art images restorations, sampled from Tab. 1(a). Users were presented with the original low-quality images alongside the restorations from Real-ESRGAN, StableSR, CoSeR, SUPIR, and InstantIR. Participants were then asked to intuitively select their preferred restoration result based on overall perceptual quality. To date, we have gathered responses from 33 participants, and the results are summarized as follows:
> > > | Real-ESRGAN | StableSR | CoSeR | SUPIR | InstantIR |
> > > | ----------------  | ----------- | -------- | ------- | ----------- |
> > > |1.21% | 22.42% | 24.24% | 4.85% | 47.27% |
> > >
> > > We believe this results highlight the effectiveness of InstantIR, especially in delivering **favored by human perception**.

---

> > > > ### Author Response · Authors · 2024-11-23
> > > > **Official Response (2/2)**
> > > >
> > > > > “I also don't see why "adjusting" the perceptual quality according to the user's experience (e.g., via a text prompt) is beneficial, if it results in a larger distortion than other methods.”
> > > >
> > > > Thank you for raising this point. The result presented in 'Official Response (1/3)' *did not involve adjustments via text prompt*, apart from the application of adaIN post-processing. As noted in our first reply, this supplementary experiment aims to demonstrate **the flexibility of InstantIR**, rather than to provide results that surpass those presented in our original manuscript.
> > > > With the reference mechanism, InstantIR offers even more flexibility in balancing the perception-distortion tradeoff beyond applying adaIN, as presented in **Tab. 2(a), Page 10**. We believe such flexibility is valuable in practice, especially given that many *users may lack the background knowledge to interpret metrics such as “PSNR” and “NIQE”*. This means they are unlikely to predefine their preferences in advance according to the plain metrics. For example, if users are unsatisfactory with an output (e.g. high NIQE), they can easily tune the results within InstantIR. In contrast, when using StableSR, users would need to reload an entirely different CoSeR model to obtain a lower NIQE score. And the same thing can happen case-by-case, the streamlined flexibility of InstantIR can enhance user experience and its adaptability in real-world applications.
> > > >
> > > > In a word, we believe InstantIR offers **competitive overall performance** in quantitative evaluations. Meanwhile, we do acknowledge that it is challenging for InstantIR to excel across all evaluation metrics simultaneously, and this should be considered as our future improvements. It is also worth noting that no existing method consistently achieves overwhelming advantages across all quantitative metrics compared with their contemporary work, especially when it comes to the tradeoff between perception and distortion. *After discussion, we concur with the reviewer's concern about claiming InstantIR as SOTA, that we should be more cautious on this point*. We have modified the related statements in **line 51, Page 1,**, and **line 532, Page10**.
> > > >
> > > > If this response address your concerns, we kindly invite the reviewer to reconsider the rating, comprehensively considering the **novelty and valuable flexibility in real-world applications** of our work, beyond quantitative evaluation metrics.

---

> > > > ### Comment · Reviewer_KzYR · 2024-11-23
> > > >
> > > > Would you consider reframing your method as "blind image **enhancement**" instead of "blind image **restoration**"? Given that the method is focused more on human experience rather than reconstruction accuracy, I believe the terminology "image restoration" is not a great fit here. This is because you are saying that the method cares more about human experience and perceived quality rather than the distortion measured by metrics which are unarguable, which may reveal minute flaws that humans cannot immediately perceive (or cannot perceive at all). In fact, while human perception is important, it may not always indicate **restoration accuracy**, which is more important in fields such as medical imaging (e.g., MRI reconstruction). So I think when a method is called "image **restoration**", human perception cannot be the only priority. It should be a priority, but **distortion must be taken into account as well**.

---

> > > > > ### Author Response · Authors · 2024-11-29
> > > > > **Official Response by Authors**
> > > > >
> > > > > Dear reviewer KzYR,
> > > > >
> > > > > We sincerely appreciate your constant time and efforts upon our work. After discussion, we believe "blind image restoration" is sufficiently good and respectfully disagree with the reframing proposal. First of all, we must apologize for the misunderstanding caused by our previous comments. It was not our intention to imply that full-reference metrics are unimportant. What we intended to convey is that, in many cases, non-reference metrics are better at reflecting image quality advantages, as the gap in PSNR can be imperceptible visually. We realize our earlier comments overlooked the importance of full-reference metrics in tasks like super resolution. We deeply apologize for our overly assertive statement and any misunderstanding it caused.
> > > > >
> > > > > After closely investigating the distortion problem, we have come into two conclusions regarding the misalignment observed in full-reference metrics:
> > > > > 1. The previewing mechanism can yield generative references deviate from the original input at early diffusion time-steps. This problem can also be seen in Fig. 3(b), Page 4;
> > > > > 2. While this deviation could be acceptable on heavily degraded inputs where information loss is severe, they can have significant impacts on public benchmarks including realsr and DrealSR that super resolution is the major object. This discrepancy can also explain why the full-reference metrics gap is widen on the 'Real-world' dataset in Tab. 1, Page 7, compared to the 'Synthetic' dataset where a mixture of degradation is applied.
> > > > >
> > > > > Intuitively, for tasks like super resolution, introducing generative references at the beginning of the generation process is unnecessary and may contribute to the observed misalignment. Following the suggestions from reviewer BMRZ, we have further refined the sampling algorithm in **Alg. 1, Page 6**, by combining idea from the restoration-guided sampling proposed in SUPIR. Specifically, we introduce a time-dependent scaler $k_{t}=(t/T)^{\tau}$ where $\tau$ is a hyper-parameter. At each sampling step, we interpolate the generative references with the original LQ latent by $k_t$. This interpolation keeps the generative references closely aligned to the target latent, avoiding distortions in the generative references previewed from large diffusion time-step. The results on realsr benchmark are summarized as follow:
> > > > >
> > > > > | Method | PSNR | SSIM | LPIPS | CLIPIQA | MANIQA | MUSIQ | NIQE |
> > > > > | ----------- | -------- | ------ | ------ | ----------- | ---------- | -------- | ------- |
> > > > > | StableSR | 24.42 | 0.7377 | 0.2547 | 0.4365 | 0.3112 | 55.24 | 6.829 |
> > > > > | SUPIR | 25.43 | 0.7176| 0.3625 | 0.3343 | 0.2407 | 41.80 | 6.174 |
> > > > > | InstantIR | 21.44 | 0.6796 | 0.3124 | 0.5099 | 0.4881 | 68.01 | 6.897 |
> > > > > | InstantIR + reference interpolation | 23.35 | 0.7291 | 0.2878 | 0.4535 | 0.4259 | 63.37 | 6.652 |
> > > > > | InstantIR + reference interpolation (512-target size) | 23.99 | 0.7391 | 0.2804 | 0.4216 | 0.3594 | 60.50 | 6.136 |
> > > > >
> > > > > Another worth-noting fact is that the test scenario we adopted for quantitative evaluation in Tab. 1(a), Page 7, downsample the outputs of InstantIR to make comparisons with other models whose outputs are constraint to 512 resolution. The downsampling will cause artifacts and downgrade the evaluation metrics. In the last row of the above table we directly generate 512 resolution outputs. We believe PSNR with values of 23.35/23.99 can be regarded at the same level as the compared methods, given the imperceptible differences from a small discrepancy of PSNR value, while other non-reference metrics improved including SSIM.

---

> > > > > > ### Comment · Reviewer_KzYR · 2024-11-29
> > > > > >
> > > > > > I appreciate the authors' dedicated efforts to improve their paper and address the reviewers' suggestions.
> > > > > > However, at this point, I believe that the paper requires too many modifications from its original form to confidently recommend acceptance.
> > > > > > The reviewer discussion process is not intended for the authors to keep adding experiments and revise their paper until it is accepted. It is intended to assess the original submission, clarify misunderstood points (and discuss them), and in some cases make minor modifications that don't significantly alter the original paper.
> > > > > >
> > > > > > Therefore, I will respectfully keep my original score.

---

> ### Author Response · Authors · 2024-11-21
> **Official Response (2/3)**
>
> > "Thus, it is not clear whether the proposed method is effective, because previous methods may also improve their perceptual quality at the expense of distortion (e.g., by tuning some hyper-parameters) and perhaps beat the proposed approach in terms of distortion at a given level of perceptual quality."
>
> This is a good point. We appreciate the contributions of these excellent work that have inspired us, and we acknowledge that some of these methods could achieve comparable performance to InstantIR with some hyper-parameter choices. However, as noted in the corresponding literature [1, 3], the authors also emphasized the **misalignment of full-reference metrics like PSNR and SSIM with human evaluation** and suggested the image quality should be prioritized. From this perspective, we believe these works have been optimized w.r.t image quality. Nevertheless, **InstantIR with adaIN still leads the image quality metrics at comparable levels of PSNR and SSIM, that is, without compromising distortion**.
>
> More importantly, while previous works could have achieved similar performance by re-training with different hyper-parameter choices, **InstantIR can directly tune this tradeoff without any fine-tuning or re-training**, as mentioned in our first reply. To this end, we believe InstantIR is more effective.
>
> > "Unfortunately, the authors do not discuss the perception-distortion tradeoff at all, which I believe is highly important given the fact that the proposed method's distortion is compromised in all data sets. Note that the authors do briefly mention this limitation in the conclusion section (the compromised distortion compared to previous methods). But again, I believe that this should be thoroughly discussed, and it should be confirmed that the proposed approach is more effective than previous ones."
>
> We appreciate your constructive suggestions that we should include the discussion of the well-known 'perception-distortion tradeoff'. We agree that this should be a critical consideration in evaluating BIR methods. We have incorporated such discussion both in ablation (**Tab. (2), Page 10; lines 463-477, Page 9; lines 485-513, Page 10**) and conclusion (**lines 527-539, Page 10**) sections. The additional experiment in Tab. 2(b) aims to provide a more comprehensive analysis of the tradeoff in our proposed method.
>
> > "The conclusion/discussion section is quite short and limited."
>
> Thank you for pointing out the problem in our conclusion section. As mentioned above in the fourth reply, we have expanded the section, please see **lines 527-539, Page 10**.
>
> > "For example, a limitation I would specify is the design choice itself, which necessitates a degradation concept perceptor (e.g., DINO). What about other image modalities besides natural images (e.g., medical images, infra-red, compressed sensing), which DINO cannot handle? How can we then train InstantIR in such circumstances? Do we need to come up with a DINO-equivalent model to operate on images in other domains? My point is that InstantIR cannot be easily generalized/applied to other image modalities."
>
> We appreciate this insightful observation and fully understand the raised concern. The choice of DINOv2 in InstantIR was driven by its excellent performance in processing natural images. See the newly added experiment in **Fig 3, Page 4**, which demonstrates the robustness of pre-trained DINOv2 against degradations. However, **it is not an irreplaceable component of InstantIR**. For other image modalities, such as those mentioned (e.g., medical images, infrared, or compressed sensing), **any pre-trained vision backbone** tailored to the specific domain tasks could be eligible. Moreover, given the efficiency our Aggregator demonstrated in references (see out-domain references in ablation study, **lines 479-483, Page 9**), only the DCP module has to be tuned in such cases.
>
> > "I believe that this should be discussed, since many practitioners are working with other image modalities where blind image restoration is a crucial task."
>
> We really appreciate this suggestion. As mentioned in the sixth reply, we have incorporated a discussion of this limitation into the conclusion section of the revised manuscript, see **lines 534-539, Page 10**.

---

> ### Author Response · Authors · 2024-11-21
> **Official Response (3/3)**
>
> > "While the proposed approach is novel, it is conceptually not that much different than previous methods (e.g., [2]). Specifically, the proposed method aims to generate samples from the posterior distribution (of natural images given a degraded image), which is an approach done in many previous works."
>
> Thank you for acknowledging the novelty of our method and for bringing up the comparison with previous methods. However, we would like to clarify that **the distinction between InstantIR and ControlNet methods lies in conceptual advancement, rather than model architecture design**. Specifically, we propose an iterative inference pipeline that **refines the generation conditions throughout the diffusion process**, as also pointed out by reviewer BMRZ. In other words, both diffusion latents and generation conditions are being updated during this process. This design ensures the generation conditions are dynamically adapted to generative prior for different inputs. In the previous method that is mentioned [4], an auxiliary pre-restoration model is employed to decompose the inference process of $p(x_{rec}|x_{lq})$ into $p(x_{rec}^{'}|x_{lq})$ and $p(x_{lq}|x_{rec}^{'})$. In contrast, In InstantIR, such a decomposition is amortized into each generation step, formally $p(x_{ref}|x_{t},x_{lq})$ and $p(x_{t-1}|x_{t}, x_{ref})$.
>
> > "Besides the empirical results, which, in my opinion, do not convince that the proposed approach is superior to previous ones (see point 1.), it is not clear why InstantIR offers any additional advantage over previous methods in terms of approximating the true sampler from the posterior."
>
> As mentioned in the first and second replies, InstantIR offers leading performance in **both quantitative metrics and visual qualities in the produced restorations**. We have highlighted the leading image scores of InstantIR in **lines 427-428, Page 8**.
>
> > "The authors do not report NIQE scores for perceptual quality, which is standard in such works (e.g., [3])."
>
> Thank you for pointing out this absence. The metrics we chose for quantitative comparison are the most widely used across our compared methods in general. We have had NIQE metrics reported in both **Tab. 1, Page 7**, and **Tab. 2, Page 10**.
>
> > "Could you please explain why the proposed method is superior to previous ones, specifically referring to the perception-distortion tradeoff? (see weakness 1). Why is it appropriate to say that InstantIR is SOTA, given the perception-distortion tradeoff?"
>
> Also, as mentioned in our first and second replies, the **PSNR and SSIM can be adjusted to the same level as other methods, while maintaining the leading performance of InstantIR in image quality metrics**. We have updated the ablation study with incorporating post-processing process like adaIN to adjust the full-reference metrics in **Tab. 2, Page 10**.
>
> > "What evidence in the paper convinces that the proposed approach is more effective than previous ones? For example, if previous methods were trained slightly differently (with different hyper-parameters), they may attain the same perceptual quality as InstantIR, but with better distortion (e.g., PSNR, SSIM)."
>
> As mentioned in our third reply. We agree that previous work could have achieved performance comparable to InstantIR with some hyper-parameters, but **InstantIR can directly be tuned on the perception-distortion tradeoff at inference time**, without a need of training for being so. Therefore, we believe it is more effective in terms of computational cost. For more evidence, please see the added ablation results in **Tab. 2, Page 10**.
>
> > "How can we apply/train InstantIR in other image modalities besides natural images?"
>
> Also mentioned in our sixth reply. We have updated the conclusion section to incorporate such discussion in **lines 534-539, Page 10**.
>
> [1] Wang, Jianyi, et al. "Exploiting diffusion prior for real-world image super-resolution." International Journal of Computer Vision (2024): 1-21.
>
> [2] Choi, Jooyoung, et al. "Perception prioritized training of diffusion models." Proceedings of the IEEE/CVF Conference on Computer Vision and Pattern Recognition. 2022.
>
> [3] Yu, Fanghua, et al. "Scaling up to excellence: Practicing model scaling for photo-realistic image restoration in the wild." Proceedings of the IEEE/CVF Conference on Computer Vision and Pattern Recognition. 2024.
>
> [4] Xinqi Lin et al., DiffBIR: Towards Blind Image Restoration with Generative Diffusion Prior, arXiv 2023

---

### Official Review · Reviewer_uPZW · 2024-10-28

**Soundness:** 2
**Presentation:** 2
**Contribution:** 2
**Rating:** 5
**Confidence:** 5

**Summary:**

The paper INSTANT-IR addresses the challenge of handling unknown test-time degradation in Blind Image Restoration (BIR) using diffusion model

The method extracts a compact representation of the input image using a pre-trained vision encoder. During each diffusion step, this representation decodes the current diffusion latent state, embedding it into the generative prior. The degraded image is encoded with this reference, which creates a robust generation condition. By observing that the variance of generative references changes with degradation intensity, the authors develop a sampling algorithm that adapts to input quality.

Experiments show that INSTANT-IR achieves state-of-the-art performance, providing excellent visual quality. By adjusting generative references with textual descriptions, it can handle extreme degradation and even perform creative restoration.

**Strengths:**

It is not easy  to conclude what's the major strengths of this paper since most of the components are pretty straightforward compared with other paper. I will give authors a chance to answer what's the  Strengths from authors point of view.

**Weaknesses:**

Here are some of the points I am not clear by reading this paper.

1. Why DINO as the model to provide compact LQ image representation. From my understanding, DINO is a self-supervised learned representation that is very good for  semantic understanding. In a sense, the dino feature is highly compact and salient so that a lot of low-level cues are missing in the DINO feature. And Image Restoration is a pretty low-level task which need to pay more on the details rather than saliency.   That's why it doesn't make sense to me that DINO is a good choice here.

2. If the Generative Reference is generated from LQ image representation, it is not clear what extra information that could bring in. In addiiton, in Line 211-212, what high-level information is missing in LQ and it is super un-clear.

3. For the case with prompt, how did author train it ? And what if the prompt and LR images are totally different like a women LR image but with a man as prompt etc. Please discuss more in this part and it is totally missing in this paper.

Overall, I didn't see any significant novelty in this paper since authors just use Dino feature and then decode it to be a reference image to guide the generation process.

**Questions:**

Please see weakness.

**Details Of Ethics Concerns:**

For portrait enhancement, not sure whether this method will introduce bias specially  when given text prompt

---

> ### Author Response · Authors · 2024-11-21
> **Official Response (1/2)**
>
> Dear Reviewer uPZW,
>
> We sincerely appreciate the reviewer's effort and constructive comments, especially regarding **the question of what additional information could be brought into the inference process**. This is an excellent angle to better understand and highlight the main idea of our method and its novelty. Below, we will address your concerns one-by-one:
>
> > "It is not easy to conclude what's the major strengths of this paper since most of the components are pretty straightforward compared with other paper. I will give authors a chance to answer what's the Strengths from authors point of view."
>
> Thank you for leaving us this opportunity. In summary, we propose an iterative inference pipeline that **refines the generation conditions throughout the diffusion process**. In other words, both diffusion latents and generation conditions are being updated during this process. This can be **analogous to the difference between VAE and diffusion model**: while both model generate from noise, diffusion model achieves superior generation capability by leveraging a stepwise inference $p(x_{t-1}|x_t)$, which is much more tractable than the one-step inference $p(x_0|x_T)$ taken by VAE. This design also ensures **the generation conditions are dynamically adapted** to generative prior in-domain for different inputs, as pointed out by reviewer BMRZ.
>
> > "Why DINO as the model to provide compact LQ image representation. From my understanding, DINO is a self-supervised learned representation that is very good for semantic understanding. In a sense, the dino feature is highly compact and salient so that a lot of low-level cues are missing in the DINO feature."
>
> This is a good point. DINO’s representation is compact with low-level cues omitted, which means it is **robust in low-level tasks where such information is absent or distorted**. To validate this, we have added a verification experiment in **Fig. 3, Page 4**. We test the pre-trained DINOv2 on ImageNet-1K classification under various degradation conditions, with the results presented as density plots in **Fig. 3(a), Page 4**. The results demonstrate DINOv2’s robustness to low-quality inputs without fine-tuning, confirming the reliability of its output representation.
>
> > "And Image Restoration is a pretty low-level task which need to pay more on the details rather than saliency. That's why it doesn't make sense to me that DINO is a good choice here."
>
> We fully agree with this perspective and would like to emphasize that **our approach aligns with this principle**. The use of DINO in our method serves solely as an **auxiliary module providing reliable information**. The high-level information in DINO representation offers semantic guidance for reference generation, effectively narrowing the probability mass to a large extent where we can sample generative references for helping the fine-details generation, as visualized in **Fig. 3(b), Page 4**. We have added this detailed explanation to **lines 204-211, Page 4**.
>
> > "If the Generative Reference is generated from LQ image representation, it is not clear what extra information that could bring in."
>
> Thank you for raising this excellent point. **The extra information directly comes from the generative model**, which can be visualized in Fig. 3(b). The generative references provide high-level features guidance in the early restoration process (larger t). As diffusion time-step decreases, the references progressively reveal finer low-level details. This behavior is consistent with the well-known characteristic of the diffusion generation process, where global structures and layouts are synthesized before local low-level details. In other words, the lack of low-level details in the references won’t affect diffusion generation in the early stage.
>
> Mathematically, Fig. 3(b) can be regarded as a visualized process of sampling from the refined posterior $p(x_0 | x_t, c_{lq})$. At the beginning (left most, t=1000), $p(x_0 | x_T, c_{lq})$ is not sufficiently constrained with probability mass spread across a large sample space, which leads to diverse sampling outputs. **With information gained from more informative diffusion latent $x_t$, the posterior $p(x_0 | x_t, c_{lq})$ is gradually narrowed towards the target mode**. We have moved the visualized generative references from the experiment section to **Fig. 3(b), Page 4**, to better explain our motivation together with the newly added DINO experiment.
>
> > "In addiiton, in Line 211-212, what high-level information is missing in LQ and it is super un-clear."
>
> Thanks for your careful review and pointing out this problem. It should have been the low-level information that is missing in the representation. We have corrected this error in **line 226, Page 5**, in our revised version.

---

> > ### Author Response · Authors · 2024-11-25
> > **Official Comment by Author**
> >
> > Dear reviewer uPZW,
> >
> > Thank you for taking the time to review our work. We would appreciate your feedback on the clarification and analysis we provided:
> >
> >  - Explanation of the strength in this work from authors' point of view.
> >  - W1: the rationale for employing DINOv2, in the context of this low-level vision task. An additional verification experiment in **lines 205-211, Page 4** and **Fig. 3, Page 4**.
> >  - W2: extra information brought from the generative prior.
> >  - W3: detailed analysis of the text-editing ability in different scenarios, including conflict/aligned semantics between text and LQ images in **Appendix A**.
> >
> > We are happy to address any further questions or concerns you may have.
> >
> > Best,
> >
> > Authors.

---

> > > ### Author Response · Authors · 2024-11-28
> > > **Follow Up**
> > >
> > > Dear reviewer uPZW,
> > >
> > > We deeply appreciate the time and effort you are dedicating to the review process. Since it is the last day of PDF revision, we would like to know whether we have addressed your further comments.
> > >
> > > If you have any additional questions or require further clarification on any aspect of our work, please do not hesitate to let us know. We are more than happy to provide any additional information or address any concerns you may have.
> > >
> > > Thank you very much for your time and attention.

---

> ### Author Response · Authors · 2024-11-21
> **Official Response (2/2)**
>
> > "For the case with prompt, how did author train it ?"
>
> We only **trained the Previewer and DCP on text-image paired data**, as presented in **lines 215-217, Pages 4-5**. The Previewer’s cross-attention operation, depicted in Fig. 2(b), is an additive combination of text $c_{txt}$ and $c_{lq}$. Also, it is worth noting that **the Aggregator is not trained on any text data and it is pure single-modal**, as presented in **lines 249-251, Page 5**. The text-editing ability InstantIR arises from both the compound transformations from both modalities in the DCP and Previewer, and the Aggregator's effectiveness in processing reference latents. We have modified **Fig. 2, Page 3**, according to reviewer y1Rv and BMRZ's suggestions to provide more details on the Previewer and Aggregator operations.
>
> > "And what if the prompt and LR images are totally different like a women LR image but with a man as prompt etc. Please discuss more in this part and it is totally missing in this paper."
>
> Thank you for this thoughtful comment. As presented in **lines 307-311, Page 6**, when there is such semantic conflict between the LQ images and the text prompts, the generated outputs may exhibit undesirable visual quality. To explore the text-editing ability of InstantIR, we conducted additional experiments by synthesizing LQ images from the ImageNet-1K validation set using the Real-ESRGAN pipeline. These images are categorized based on their DINOv2 classification scores. Our analysis reveals that as classification scores increases—indicating richer semantics in the DINO representation—the flexibility of text-based editing decreases. At moderate classification scores, **the two modalities exert a balanced influence, and semantic conflicts can result in unpleasant outcomes**. We have included visual examples and corresponding analysis of this experiment in **Appendix A**.
>
> > "Overall, I didn't see any significant novelty in this paper since authors just use Dino feature and then decode it to be a reference image to guide the generation process."
>
> We appreciate the reviewer's concern about the novelty of our proposed method. In conclusion, our main idea can be summarized into: 1) we highlight the representation from pre-trained **DINOv2 as a reliable condition to sample generative references**; 2) as diffusion process iteratively updates the latent states $x_{t}$ to better sample from $p(x_0)$, we **iteratively update the generation conditions** by invoking generative references at each step, as we mentioned in the above replies.

---

### Official Review · Reviewer_BMRZ · 2024-11-04

**Soundness:** 3
**Presentation:** 2
**Contribution:** 3
**Rating:** 6
**Confidence:** 5

**Summary:**

The paper presents a novel approach to Blind Image Restoration (BIR) called Instant-reference Image Restoration (INSTANTIR), which addresses the challenge of model generalization under unknown degradation conditions. The authors propose a diffusion-based method that dynamically adjusts the generation conditions during inference by utilizing prior knowledge extracted from a pre-trained vision encoder.  Overall, the paper contributes a significant advancement in the field of image restoration by effectively integrating generative models with adaptive sampling strategies, demonstrating robust results across various degradation levels.

**Strengths:**

1. The design of the proposed method seems to be reasonable. In fact, different images should be enhanced with different generation ability, previous works do not address this issue. Meanwhile, this work introduces a solution for this important issue, which is important.
2.  The flexibility of the proposed method is verified, which may contribute to the usage of the diffusion model for image restoration.
3.  The paper summarizes the advantages of INSTANTIR and points out possible future research directions, including improving the interaction between generated priors and conditions, which is important in this topic.

**Weaknesses:**

1. The figure seems to be naive, I need to read the method carefully to get the key idea of the proposed method. Can authors provide a more detailed figure for both Sec.3.2 and Sec.3.3 to make it more clear?
2. In Sec.3.3, the authors claim "By employing a text-guided Previewer, we can generate diverse restoration variations with compound semantics from both modalities. However, these variation samples can conflict with the original input, making them ineligible as generative references." However, the authors do not provide visual analysis or references to support this opinion.
3. It seems the quality of the reference produced by the Previewer seems to be important for the final restoration effect, can authors provide deep analysis for the effect of the reference quality beyond simply visual results?
4. Since there are two diffusion models in the architecture, the computation cost and efficacy of the proposed method should be discussed.

**Questions:**

Please see the above Weaknesses.

---

> ### Author Response · Authors · 2024-11-21
> **Official Response**
>
> Dear reviewer BMRZ,
>
> We sincerely appreciate the reviewer’s efforts, and the recognition of **the importance of adapting generation conditions for different inputs using generative prior**. Below we will address your other concerns in detail one-by-one.
>
> > "The figure seems to be naive, I need to read the method carefully to get the key idea of the proposed method. Can authors provide a more detailed figure for both Sec.3.2 and Sec.3.3 to make it more clear?"
>
> Thanks for your feedback regarding Fig. 2, the figure has been re-designed according to your suggestion, please see **Fig. 2, Page 3**. We separate Fig. 2 into 3 sub-figures with different focuses. Specifically, Fig. 2(a) illustrates the overall pipeline, stressing out the core previewing mechanism. Meanwhile Fig. 2(b) and Fig. 2(c) give more details on the operations in the Previewer and Aggregator module. The caption is also adapted according to the content change, see **lines 120-128, Page 3**. For Sec 3.3, the main idea and procedure are summarized in **Alg. 1, Page 6**.
>
> > "In Sec.3.3, the authors claim 'By employing a text-guided Previewer, we can generate diverse restoration variations with compound semantics from both modalities. However, these variation samples can conflict with the original input, making them ineligible as generative references.' However, the authors do not provide visual analysis or references to support this opinion."
>
> Thank you for pointing out this problem. To substantiate our claim, we conducted additional experiments by synthesizing LQ images from the ImageNet-1K validation set using the Real-ESRGAN pipeline [1]. These images are categorized based on their DINOv2 classification scores. It turns out that **as classification scores increases—indicating richer high-level semantics in the DINO representation—the flexibility of text-based editing decreases**. At moderate classification scores of 40% ~ 60%, the two modalities exert a balanced influence, and **semantic conflicts can result in unpleasant outcomes in such situations**. We have included visual examples and detailed analysis of this experiment in **Appendix A**.
>
> > "It seems the quality of the reference produced by the Previewer seems to be important for the final restoration effect, can authors provide deep analysis for the effect of the reference quality beyond simply visual results?"
>
> Thank you for highlighting this crucial aspect of our work. Indeed, the quality of references significantly affects the performance of InstantIR. To provide a more in-depth analysis, we have **added an ablation study using different sources of references** with progressively increasing quality: LQ image, HQ image, DDIM mean, unconditional restoration preview, restoration preview with DCP and additionally with text prompt. The results are summarized in **Tab. 2(a), Page 10**. As discussed in **lines 463-477, Page 9**. Using more constrained references—either those generated by our distilled Previewer or those with additional conditions like DCP and text prompts—leads to a continuous improvement in non-reference metrics that evaluate image quality.
>
> Ablation study of different sources of references, the generative references have an ascending order of quality. Best results are marked as **bold**.
> | Reference | PSNR | SSIM | LPIPS | CLIPIQA | MANIQA | MUSIQ | NIQE |
> | ----------- | -------- | ------ | ------ | ----------- | ---------- | -------- | ------- |
> | LQ Image | **21.36** | **0.6417** | 0.4950 | 0.2415 | 0.2025 | 33.39 | 8.049 |
> | HQ Image | 16.86 | 0.5791 | **0.3728** | 0.5078 | 0.3892 | 65.72 | 5.139 |
> | DDIM Mean | 21.10 | 0.6066 | 0.4000 | 0.4515 | 0.3727 | 60.93 | 5.819 |
> | Unconditional Preview | 20.94 | 0.6108 | 0.3787 | 0.5023 | 0.4052 | 65.71 | 5.168 |
> | Preview + DCP | 18.77 | 0.5514 | 0.3933 | 0.5941 | 0.4687 | 70.45 | **4.658** |
> | Preview + DCP + text | 18.01 | 0.5202 | 0.4065 | **0.6489** | **0.5112** | **72.32** | 4.669 |
>
> > "Since there are two diffusion models in the architecture, the computation cost and efficacy of the proposed method should be discussed."
>
> We acknowledge the reviewer’s concern regarding the computational cost of our method. As the Previewer model will take another NFE per step, **we only use 30 DDIM** steps as we presented in **line 368, Page 7**, compared to the 50–100 steps samplers employed by other diffusion-based methods. It is worth noting that larger sampling steps will yield finer details, which is crucial for the BIR task. All the results from diffusion-based models presented in Tab. 1 (Page 7) **take approximately 30 seconds per image restoration on a single V100 GPU**. We have revised **lines 421–423, Page 8,** to clarify this configuration.
>
> In terms of storage, the Previewer is trained by applying LoRA to SDXL, which means we only need one U-Net saved in memory, as detailed in **lines 358-360, Page 7**.
>
> [1] Xintao Wang et al., Real-ESRGAN: Training Real-World Blind Super-Resolution with Pure Synthetic Data, ICCV 2021

---

> > ### Author Response · Authors · 2024-11-25
> > **Official Comment by Authors**
> >
> > Dear reviewer BMRZ,
> >
> > Thank you for taking the time to review our work. We would appreciate your feedback on the clarification and analysis we provided:
> >
> >  - W1: the unclear illustration of naive **Fig. 2**, which has been redesigned and redrawn in a detailed way.
> >  - W2: visual comparisons and detailed analysis of semantic conflicts in text-prompt and LQ images in **Appendix A**.
> >  - W3: detailed analysis of how the references' quality affect restoration in **lines 465-480, Page 9** and **Tab 2(a), Page 10**.
> >  - W4: clarify the computational cost used in quantitative evaluation in **lines 421-422, Page 8**.
> >
> > We are happy to address any further questions or concerns you may have.
> >
> > Best,
> >
> > Authors.

---

> > > ### Comment · Reviewer_BMRZ · 2024-11-26
> > >
> > > Thanks for the authors' response, the response has partially addressed my concerns. Since there are similar designs (sampling strategy) to keep a balance between generation and restoration in SuperIR(CVPR 2024) and DiffBIR(ECCV 2024), can authors combine these methods with the method in this paper? A thorough discussion of these similar designs for the same issues can be provided.

---

> ### Author Response · Authors · 2024-11-27
> **Official Comment by Authors**
>
> Dear reviewer BMRZ,
>
> Thanks for your response and extra efforts devoted to our work. We are happy to address your further concerns.
>
> > “Since there are similar designs (sampling strategy) to keep a balance between generation and restoration in SuperIR(CVPR 2024) and DiffBIR(ECCV 2024), can authors combine these methods with the method in this paper?”
>
> Thanks you for raising this insightful question. Balancing generative capacity and fidelity to the input LQ image is a crucial aspect of developing generative-based image restoration models. Among the models we compared, DiffBIR, SUPIR and StableSR each implements unique sampling strategies to achieve quality-fidelity balancing with different approaches.
>
> As a core component of DiffBIR pipeline, a pre-trained IR module not only provides diffusion sampling conditions, but also is used to balance quality-fidelity. Similar to our approach, DiffBIR also retrieves the DDIM mean $\hat{z}_0^t$ at each diffusion time-step $t$. $\hat{z}_0^t$ is decoded into pixel space using SD-VAE to obtain $\mathcal{D}(\hat{z}_0^t)$. This intermediate output is used to calculate an MSE loss with the IR module output, which typically is high PSNR with sub-optimal perceptual quality, and the gradient is back-propagated to update $\hat{z}_0^t$. Since we do not have a pre-processing IR model, this strategy cannot be adopted into our pipeline. However, our approach is more efficient in two aspects: 1) we save both memory and computational cost from incorporating a pre-processing model; 2) we directly process the previews in latent space using the Aggregator, eliminating the computational cost in repeatedly calling the SD-VAE decoder.
>
> StableSR adopts the Controllable Feature Wrapping (CFW) module introduced in CodeFormer [1]. Specifically, the SD-VAE is tuned on LQ images. The encoder is optimized for degradation robustness, ensuring the LQ latents are close to the corresponding HQ latents. On the other hand, residual connections from the LQ encoder features are added to decoder to preserve original information. Since StableSR is trained on SD-2-1, the available VAE checkpoint is not compatible with the SDXL model in InstantIR. However, we believe integrating this strategy into our pipeline could potentially enhance the quality-fidelity balancing.
>
> The VAE is also fine-turned in SUPIR [2]. Unlike StableSR, SUPIR does not adjust the decoder for applying CFW module. SUPIR utilizes the degradation robust encoder as an initial restoration $z_{lq}^{'}$. At each diffusion step, the diffusion mean is interpolated with $z_{lq}^{'}$ using a time-dependent scaler $k_{t}=(t/T)^{\tau}$. Smaller $\tau$ corresponds to larger $k_{t}$, making the interpolated diffusion mean closer to $z_{lq}^{'}$.
>
> In our Adaptive Restoration (AdaRes), the scaling factor $\delta$ is also time-dependent as $k_t$. However, $k_t$ depends only on time-step $t$ and $\delta$ in AdaRes is adaptive to different inputs, offering additional flexibility. As SUPIR is built on SDXL its encoder can be incorporated into InstantIR pipeline. To make a comprehensive comparison, we consider three implementations of adopting both restoration-guided sampling and AdaRes:
> 1. InstantIR with SUPIR’s VAE encoder;
> 2. InstantIR with SUPIR restoration-guided sampling;
> 3. InstantIR with interpolation in the Aggregator inputs.
>
> | Sampling Strategy | PSNR | SSIM | LPIPS | CLIPIQA | MANIQA | MUSIQ | NIQE |
> | ----------- | -------- | ------ | ------ | ----------- | ---------- | -------- | ------- |
> | 0. InstantIR (baseline) | 18.77 | 0.5514 | 0.3933 | 0.5941 | 0.4687 | **70.45** | **4.658** |
> | 1. InstantIR + SUPIR encoder | 19.62 | 0.5877 | 0.4238 | 0.5675 | 0.4627 | 69.31 | 5.319 |
> | 2. InstantIR + SUPIR sampling | 20.04 | 0.5941 | **0.3788** | 0.5631 | 0.4514 | 69.09 | 5.036 |
> | 2.1 InstantIR + SUPIR encoder + SUPIR sampling | **21.69** | **0.6358** | 0.4138 | 0.4352 | 0.3643 | 61.67 | 5.696 |
> | 3. InstantIR + SUPIR encoder + reference interpolate | 19.81 | 0.5608 | 0.4372 | **0.5977** | **0.4905** | 69.96 | 5.472 |
>
> Because the tuned encoder has a latent space different from SDXL, implementations 1 & 2 both exhibit degraded performance with the SUPIR’s encoder (comparing lines 2&3 and 4&5). Comparing lines 2&4, the restoration-guided sampling algorithm in SUPIR can be seamlessly integrated into InstantIR providing improved full-reference metrics and slightly declined non-reference metrics. In implementation 3 we try to interpolate the LQ latents before sending to the Aggregator. This implementation borrows the idea of SUPIR's algorithm but is more tailored for InstantIR’s pipeline. Both full-reference and non-reference metrics improved (comparing lines 2&6), demonstrating the efficiency and flexibility of the proposed reference mechanism.
>
> [1] Zhou, Shangchen, et al. "Towards robust blind face restoration with codebook lookup transformer." Advances in Neural Information Processing Systems 35 (2022): 30599-30611.

---

> > ### Comment · Reviewer_BMRZ · 2024-11-28
> >
> > Thanks for your response, the rebuttal has addressed my concerns, and I strongly suggest authors to add the above discussions in the updated version of this paper. Moreover, the main figure can be further improved to look more understandable, and more visualizations about how the proposed method process different degradations can be provided if possible (such as T-SNE). Considering the rebuttal and the quality of this paper, I will maintain the positive score and raise the confidence.

---

### Official Review · Reviewer_y1Rv · 2024-11-04

**Soundness:** 3
**Presentation:** 3
**Contribution:** 2
**Rating:** 5
**Confidence:** 4

**Summary:**

A good baseline for blind image restoration. Although the performance is not good, the editing ability is good in some real-world applications.

**Strengths:**

Good theoretical analysis and well written manuscript. Some experimental results are used to prove the effectiveness of design.

**Weaknesses:**

The performance on blind image restoration is not so good as other methods. The pipeline is simple and not novel.

**Questions:**

1. The details of Instant Restoration Previewer is not well written. The authors only know that the previewer is a diffusion model. The same things happened on aggregator and DCP. In other word, Fig. 2 need to be redrawn in a detailed way.
2. Do the authors train DINOv2 for DCP from scratch? Also how to distill the diffusion model fro Previewer from a pre-trained DINOv2? It is confusing as it is said in Sec. 4.1.
3. As a workflow of blind image restoration, it contains deblur, SR, denoise and dejpeg. What is the differences between all-in-one and it? Can it also be evaluated on the all-in-one setting, such as Perceive-IR, PromptIR or ProRes?
4. The performance of the proposed method is not superior than other compared methods. The authors should press more on the essential keys of this paper.
5. The ability of editing LQ images seems to be derived from DiNOv2. The authors should provide more examples of editing scenario and discuss it in details.

Scores can be raised upon replies.

---

> ### Author Response · Authors · 2024-11-21
> **Official Response (1/2)**
>
> Dear reviewer y1Rv,
>
> We appreciate the reviewer’s efforts and recognition of the significance of text-editing ability in real-world image restoration application. Below we will address your concerns in detail.
>
> >  "The pipeline is simple and not novel."
>
> As pointed out by reviewer BMRZ, our method adapts the generation conditions according to different inputs, which is a novel pipeline including:
> 1. We highlight the use of image recognition models like DINOv2 in blind image restoration. Its high-level representations are robust to various degradations, which is a reliable source of information to cast conditional sampling the generative references.
> 2. Built on the Diffusion Model, we propose a novel restoration pipeline that **iteratively updates both diffusion latents and the generation condition**. The generation condition (low-quality image) is actively aligned with the refined posterior $p(z_0 | z_t, c_{lq})$ in this process.
>
> We believe these contributions underscore the originality and innovation of our proposed method. We have added a verification experiment in Page 4 to better support point 1.
>
> > "The details of Instant Restoration Previewer is not well written. The authors only know that the previewer is a diffusion model. The same things happened on aggregator and DCP. In other word, Fig. 2 need to be redrawn in a detailed way."
>
> Thank you for pointing out the problem of Fig. 2 and the need for additional detail. We have redesigned **Fig. 2 (Page 3)** and divided it into three sub-figures to enhance clarity. Specifically:
> - Fig. 2(a) illustrates the overall pipeline of the proposed method, highlighting the key previewing mechanism, which forms the core of our method's advantage.
> - Fig. 2(b) provides a detailed block structure of the Previewer model.
> - Fig. 2(c) elaborates on the key operations of the Aggregator.
>
> Also, we have revised the figure caption in **lines 120-128, Page 3**, to provide a more comprehensive explanation of each component. We hope this revised figure effectively addresses your concerns.
>
> >  "Do the authors train DINOv2 for DCP from scratch?"
>
> No, we use **the off-the-shelf DINOv2 model and keep it frozen** throughout the entire training process, as described in **lines 205-212, Page 4**. For efficiency, we **only trained the Resampler layer** connecting DINOv2 and SDXL. We acknowledge the confusion caused by our earlier description, as we only mention the SDXL is frozen in Sec. 4.1. We have revised **lines 211-212, Page 4**, to be more explicit on this point.
>
> > "Also how to distill the diffusion model fro Previewer from a pre-trained DINOv2? It is confusing as it is said in Sec. 4.1."
>
> Thank you for pointing out this problem. To clarify, the consistency distillation is the training procedure for **Previewer (which is a diffusion model) instead of DCP (which is related to DINOv2)**. Specifically, this procedure distills the Previewer into a one-step decoder that directly sample from $p(x_0 | x_t, c_{lq})$, which typically requires iterative inference in a vanilla diffusion model. Details of Previewer’s training are presented in **lines 225-239, Page 5**. We have revised **lines 323 & 356, Page 6-7**, to eliminate ambiguity.
>
> > "As a workflow of blind image restoration, it contains deblur, SR, denoise and dejpeg. What is the differences between all-in-one and it? Can it also be evaluated on the all-in-one setting, such as Perceive-IR, PromptIR or ProRes?"
>
> I am afraid not. InstantIR cannot be evaluated under the settings like Perceive-IR, PromptIR or ProRes. As a BIR model, InstantIR is capable of both deblur, SR, denoise and deJPEG whether individually or jointly. *All-in-one restoration is another classic low-level vision task with quite a different task setting* – It puts an emphasis on the versatility across degradations, often additionally includes some special degradations like rain, haze or low-light. However, the pattern (parameter) of each degradation is known and fixed. For example, Perceive-IR [4] and PromptIR [5] only consider denoising Gaussian noise with $\sigma=\{10,15,20\}$. In contrast, *BIR focuses on addressing the complex combinations of elementary degradations, including blur, LR, noisy and JPEG. It emphasizes the unknown patterns encountered at test time*. For example, the widely adopted Real-ESRGAN [1] degradation synthesis pipeline uses random parameters for constructing noise kernels, which leads to diverse noisy patterns in both training and testing. *Generally, comparisons of models across these tasks are not considered as standard practice in the literature, since it is difficult to establish a fair comparison*. In Tab. 1 InstantIR is evaluated under a synthesized multi-degradation setting, which requires **deblur, SR, denoise and deJPEG simultaneously** following [1]. We have revised **lines 410-412, Page 8**, to clarify the restoration task of our synthesized test set.

---

> > ### Author Response · Authors · 2024-11-25
> > **Official Comment by Authors**
> >
> > Dear reviewer y1Rv,
> >
> > Thank you for taking the time to review our work. We would appreciate your feedback on the clarification and analysis we provided:
> >
> >  - W1: the pipeline is simple and not novel.
> >  - W1 & Q4: The performance of the proposed method is not superior than other compared methods.
> >  - Q1: details of the Previewer, Aggregator illustrated in Fig. 2.
> >  - Q2: training details of DCP and Previewer distillation.
> >  - Q3: difference to all-in-one restoration task.
> >  - Q5: detailed analysis of text-editing ability in different scenarios.
> >
> > We are happy to address any further questions or concerns you may have.
> >
> > Best,
> >
> > Authors.

---

> ### Author Response · Authors · 2024-11-21
> **Official Response (2/2)**
>
> > "The performance of the proposed method is not superior than other compared methods. The authors should press more on the essential keys of this paper."
>
> Thanks for raising this point. We would like to argue that the **human perspective quality should be prioritized in BIR**. In **Tab. 1, Page 7** we present the quantitative comparisons against other latest BIR models. Note that in terms of image quality metrics MANIQA, CLIPIQA, MUSIQ, NIQE, InstantIR achieves an **average ranking of 1.48** across the compared models, which is a **leading performance** compared to other SOTA baselines: SUPIR (#3.31), CoSeR (#1.69), StableSR(#3.94), Real-ESRGAN(#5), BSRGAN(#5.88). We’ve highlighted the results in **lines 426-427, Page 8**.
>
> As for the non-reference metrics, we can incorporate post-processing procedures like adaptive image normalization (adaIN) in [6]. We have added an additional ablation study of combining  adaIN and Alg. 1 in **Tab. 2(b), Page 10**, as a direct comparison to the result in Tab.1(a)’s Real-world benchmark. The **PSNR and SSIM can be adjusted to the same level as other methods, while maintaining the leading performance of InstantIR in image quality assessments**. While this adjustment can generalize across entries in Tab. 1, **we opt to retain the original results since we prioritize human perspective quality in image restoration**.
>
> InstantIR adjusted with color correction, best scores marked as **bold**:
>
> | Model | PSNR | SSIM | LPIPS | CLIPIQA | MANIQA | MUSIQ | NIQE |
> | ------- | ------- | ------ | ------- | ---------- | ---------- | --------- | ------ |
> | BSRGAN | 26.38 | 0.7651 | 0.4120 | 0.3151 | 0.2147 | 28.58 | 9.528 |
> | Real-ESRGAN | **27.29** | **0.7894** | 0.4173 | 0.2532 | 0.2398 | 25.66 | 8.561 |
> | StableSR | 26.40 | 0.7721 | **0.2597** | 0.4501 | 0.2947 | 48.79 | 7.724 |
> | CoSeR | 25.59 | 0.7402 | 0.2788 | **0.5809** | 0.3941 | 60.51 | 6.514 |
> | SUPIR | 26.41 | 0.7358 | 0.3639 | 0.3869 | 0.2721 | 42.72 | 8.550 |
> | **InstantIR (w/o adaIN)** | 21.75 | 0.6766 | 0.3686 | 0.5401 | **0.4819** | **65.32** | **6.064** |
> | **InstantIR (w/ adaIN)** | 24.51 | 0.7102 | 0.3558 | 0.5319 | 0.4672 | 64.56 | 7.997 |
>
> Additionally, we conducted an anonymous user study during the author response period. Participants were presented with the original low-quality images, and asked to intuitively select their preferred restoration results. We have gathered 33 responses summarized as follows:
> | Real-ESRGAN | StableSR | CoSeR | SUPIR | InstantIR |
> | ----------------  | ----------- | -------- | ------- | ----------- |
> |1.21% | 22.42% | 24.24% | 4.85% | **47.27%** |
>
> We believe this results highlight the effectiveness of InstantIR, especially in delivering **human favored quality**.
>
> > "The ability of editing LQ images seems to be derived from DiNOv2. The authors should provide more examples of editing scenario and discuss it in details."
>
> Yes, the text-editing ability is **partly attributed to DINOv2**. The compact representation from DINOv2 provides high-level features of the original input, which indeed plays a role in the editing process. However, it is important to note that *DINOv2's compact representation loses information to different extent, which leaves spaces for the injection of semantics from text prompts*. In DCP we introduce an additional cross-attention layer for processing DINO representation in parallel to text modality. Since the output states are added together, text descriptions can complement/modify the high-level features absent in DINOv2’s representation.
>
> To validate this, we synthesize LQ images from the ImageNet-1K validation set using the Real-ESRGAN pipeline [1]. These images were categorized based on their DINOv2 classification scores. **As the classification scores decrease, indicating more high-level information absent in DINO representation, the flexibility of text-editing increases**. We have added visual examples and detailed analysis of this experiment in **Appendix A** to discuss the creative restoration ability of InstantIR in detail.
>
> [1] Xintao Wang et al., Real-ESRGAN: Training Real-World Blind Super-Resolution with Pure Synthetic Data, ICCV 2021
>
> [2] Yu, Fanghua, et al. "Scaling up to excellence: Practicing model scaling for photo-realistic image restoration in the wild." Proceedings of the IEEE/CVF Conference on Computer Vision and Pattern Recognition. 2024.
>
> [3] Lin, Xinqi, et al. "Diffbir: Towards blind image restoration with generative diffusion prior." arXiv preprint arXiv:2308.15070(2023).
>
> [4] Zhang, Xu, et al. "Perceive-IR: Learning to Perceive Degradation Better for All-in-One Image Restoration." arXiv preprint arXiv:2408.15994 (2024).
>
> [5] Potlapalli, Vaishnav, et al. "Promptir: Prompting for all-in-one image restoration." Advances in Neural Information Processing Systems 36 (2024).
>
> [6] Wang, Jianyi, et al. "Exploiting diffusion prior for real-world image super-resolution." International Journal of Computer Vision (2024): 1-21.

---

> > ### Comment · Reviewer_y1Rv · 2024-11-26
> >
> > Thanks for replying to all my questions. Most of my concerns are solved with the authors detailed rebuttal.
> >
> > The only concern in this manuscript is **the misalignment PSNR** between INSTANTIR and other compared methods. I may not agree with the authors regarding to the nonsense of PSNR and much more important on non-reference metrics.
> >
> > However, it is true that visual results show less differences **even with a huge PSNR gap**. I doubt whether the samples shown here is good or all the test results are like these. I agree that SUPIR and StableSR show a gap in PSNR, but not so huge.
> >
> > As other reviewers said, if you view INSTANTIR as for image restoration, I would insist that INSTANTIR is not perfect as the authors cannot explain the huge gap and this phenomenon is not solved and **even worse** than previous works. INSTANTIR  is more like a way to better enhance human-centric visual quality, not simply BIR.
> >
> > BTW, for the Appendix A, I wonder whether there should be 'woman with classes' or 'woman with glasses' in Fig. 9.

---

> ### Author Response · Authors · 2024-11-27
> **Official Comment by Authors**
>
> Dear reviewer y1Rv,
>
> We sincerely appreciate your additional efforts and thoughtful comments on our work. We are pleased to know that most of your concerns have been resolved, with the exception of the misalignment regarding full-reference metrics like PSNR. First, we must apologize for the misunderstanding caused by our previous comments. It was not our intention to imply that full-reference metrics are unimportant. What we intended to convey is that, in many cases, non-reference metrics are better at reflecting image quality advantages, as the gap in PSNR can be imperceptible visually. We realize our earlier comments overlooked the importance of full-reference metrics in tasks like super resolution. We deeply apologize for our overly assertive statement and any misunderstanding it caused.
>
> After closely investigating the problem, we have come into two conclusions regarding the misalignment observed in full-reference metrics:
> 1. The previewing mechanism can yield generative references deviate from the original input at early diffusion time-steps. This problem can also be seen in Fig. 3(b), Page 4;
> 2. While this deviation could be acceptable on heavily degraded inputs where information loss is severe, they can have significant impacts on public benchmarks including realsr and DrealSR that super resolution is the major object. This discrepancy can also explain why the full-reference metrics gap is widen on the 'Real-world' dataset in Tab. 1, Page 7, compared to the 'Synthetic' dataset where a mixture of degradation is applied.
>
> Intuitively, for tasks like super resolution, introducing generative references at the beginning of the generation process is unnecessary and may contribute to the observed misalignment. Following the suggestions from reviewer BMRZ, we have further refined the sampling algorithm in **Alg. 1, Page 6**, by combining idea from the restoration-guided sampling proposed in SUPIR. Specifically, we introduce a time-dependent scaler $k_{t}=(t/T)^{\tau}$ where $\tau$ is a hyper-parameter. At each sampling step, we interpolate the generative references with the original LQ latent by $k_t$. This interpolation keeps the generative references closely aligned to the target latent, avoiding distortions in the generative references previewed from large diffusion time-step. The results on realsr benchmark are summarized as follow:
>
> | Method | PSNR | SSIM | LPIPS | CLIPIQA | MANIQA | MUSIQ | NIQE |
> | ----------- | -------- | ------ | ------ | ----------- | ---------- | -------- | ------- |
> | StableSR | 24.42 | 0.7377 | 0.2547 | 0.4365 | 0.3112 | 55.24 | 6.829 |
> | SUPIR | 25.43 | 0.7176| 0.3625 | 0.3343 | 0.2407 | 41.80 | 6.174 |
> | InstantIR | 21.44 | 0.6796 | 0.3124 | 0.5099 | 0.4881 | 68.01 | 6.897 |
> | InstantIR + reference interpolation | 23.35 | 0.7291 | 0.2878 | 0.4535 | 0.4259 | 63.37 | 6.652 |
> | InstantIR + reference interpolation (512-target size) | 23.99 | 0.7391 | 0.2804 | 0.4216 | 0.3594 | 60.50 | 6.136 |
>
> Another worth-noting fact is that the test scenarios we designed for Tab. 1(a), Page 7, downsample the outputs of InstantIR to make comparisons with other models whose outputs are constraint to 512 resolution. The downsampling will cause artifacts and downgrade the evaluation metrics. In the last row of the above table we directly generate 512 resolution outputs. We believe PSNR with values of 23.35/23.99 can be regarded at the same level as the compared methods, given the imperceptible differences from a small discrepancy of PSNR value, while other non-reference metrics improved including SSIM.
>
> > "BTW, for the Appendix A, I wonder whether there should be 'woman with classes' or 'woman with glasses' in Fig. 9."
>
> Thank you for your careful inspection, this mistake has been eliminated.

---

> > ### Author Response · Authors · 2024-11-29
> > **Follow Up**
> >
> > Dear reviewer y1Rv,
> >
> > We deeply appreciate the extra time and efforts you have devoted to our work. Since it is the PDF revision of manuscript will due within 24 hours, we would like to hear your feedback regarding our refined sampling algorithm to improve distortion and PSNR.
> >
> > If you have any additional questions or require further clarification on any aspect of our work, please do not hesitate to let us know. We are more than happy to provide any additional information or address any concerns you may have.
> >
> > Thank you very much for your time and attention.

---

### Author Response · Authors · 2024-11-21
**Summary of Rebuttal Process**

We extend our heartfelt thanks to all reviewers for their time, efforts, and insightful feedback. We are greatly encouraged by the reviewer’s recognition on the **significance of text-guided restoration in real-world application** (y1Rv), **the importance of adapting conditions to different inputs** (BMRZ), and **the novelty of the proposed method in comparison to previous work** (KzYR).
During the author response period, we have considered the constructive suggestions provided and made several significant improvements our manuscript, including:
- A more detailed illustration of our methods, encompassing both an overview of the method and explanations in key operations.
- A verification experiment to clarify the rationale for employing DINOv2, an image recognition model, in the context of this low-level vision task.
- A deeper analysis of how both in-domain and out-domain references contribute to blind image restoration in our method.
- A comprehensive discussion on the perception-distortion tradeoff observed in our proposed method, along with justification for its claim as a state-of-the-art approach.

All changes have been **marked in red in the revised paper**, reflecting substantial improvements to our work. Once again, we sincerely thank the reviewers for their thoughtful suggestions and valuable contributions to enhancing our paper.

---

### Meta-Review · Area_Chair_x7nx · 2024-12-16

**Metareview:**

This paper proposes InstantIR, a new method for blind image restoration (BIR) that uses a diffusion-based approach. The authors claim their method achieves state-of-the-art performance in BIR, offering high perceptual quality. They also claim their method is flexible and can be used to restore extremely degraded images.

Strengths:
- The paper is generally well-written and easy to understand.
- The method is able to restore extremely degraded images.
- The method is flexible and can be used to generate a variety of outputs.

Weaknesses:
- The performance of the method is not superior to other compared methods in terms of PSNR and SSIM.
- The pipeline doesn't offer significant novelty, the contribution is very limited.
- The authors do not provide enough examples of editing scenarios.
- Discussion about important topics such as the perception-distortion tradeoff is very limited.

In its current presentation, the weakness overweight the strengths.  Despite revisions, some reviewers remain unconvinced of the method's practical value and impact.

**Additional Comments On Reviewer Discussion:**

During the rebuttal period, the reviewers raised several concerns about the paper. These concerns included the following:

- The performance of the method is not superior to other compared methods.
- The pipeline is simple and not novel.
- The authors do not provide enough details on the Instant Restoration Previewer, aggregator, and DCP.
- The authors do not provide enough examples of editing scenarios.
- The authors do not discuss the perception-distortion tradeoff.

The authors responded to these concerns by providing additional details on their method and by conducting additional experiments.
However, the authors have not adequately addressed all of the reviewers' concerns. In particular, the authors have not been able to demonstrate that their method is superior to other compared methods.

---

### Decision · Program_Chairs · 2025-01-22

Reject